# Reinforcement regulates timing variability in thalamus

**Jing Wang[1,2]\*, Eghbal Hosseini[2], Nicolas Meirhaeghe[3], Adam Akkad[2], Mehrdad Jazayeri[2]\***

[1]Department of Bioengineering, University of Missouri, Columbia, United States; [2]McGovern Institute for Brain Research, Department of Brain and Cognitive Sciences, Massachusetts Institute of Technology, Cambridge, United States; [3]Harvard-MIT Division of Health Sciences and Technology, Cambridge, United States

**Abstract** Learning reduces variability but variability can facilitate learning. This paradoxical relationship has made it challenging to tease apart sources of variability that degrade performance from those that improve it. We tackled this question in a context-dependent timing task requiring humans and monkeys to flexibly produce different time intervals with different effectors. We identified two opposing factors contributing to timing variability: slow memory fluctuation that degrades performance and reward-dependent exploratory behavior that improves performance. Signatures of these opposing factors were evident across populations of neurons in the dorsomedial frontal cortex (DMFC), DMFC-projecting neurons in the ventrolateral thalamus, and putative target of DMFC in the caudate. However, only in the thalamus were the performance-optimizing regulation of variability aligned to the slow performance-degrading memory fluctuations. These findings reveal how variability caused by exploratory behavior might help to mitigate other undesirable sources of variability and highlight a potential role for thalamocortical projections in this process.

**\*For correspondence:**
jingwang.physics@gmail.com (JW);
mjaz@mit.edu (MJ)

**Competing interests:** The authors declare that no competing interests exist.

## Introduction

While interacting with a dynamic and uncertain environment, humans and animals routinely rely on trial-and-error learning strategies to optimize their behavior. This type of learning is most rigorously formulated within the framework of reinforcement learning (RL). The key idea behind RL is to compute a value function for all possible actions based on previous outcomes, and use those action-outcome relationships to guide future actions (*Sutton and Barto, 1998*). RL has been extremely valuable in explaining behavior when an agent has to choose among a small number of discrete options, such as classic multi-armed bandit problems (*Daw et al., 2005*). However, learning through trial and error is also critical when the action space is large and continuous (*Dhawale et al., 2017*). A case in point is when the agent has to adjust the kinematics of its movements based on binary or scalar (unsigned) feedback (*Izawa and Shadmehr, 2011*; *Shmuelof et al., 2012a*; *Dam et al., 2013*; *Wu et al., 2014*; *Nikooyan and Ahmed, 2015*; *Pekny et al., 2015*; *Chen et al., 2017*). In this case, classic RL is not feasible as it requires the agent to compute the value function over an infinitude of parameters (e.g. all the kinematic variables associated with a reach and grasp movement). Moreover, in a continuous state space, errors, no matter how small, interfere with the agent's ability to update action-outcome relationships.

How do humans and animals employ trial-and-error learning strategies when the space of possibilities is continuous? One intriguing hypothesis is that the brain directly regulates variability to facilitate learning (*Kao et al., 2005*; *Ölveczky et al., 2005*; *Tumer and Brainard, 2007*; *Huang et al., 2011*). The basic idea is that the agent reduces variability when a trial yields reward (i.e. exploit

previously rewarded action) and increases variability in the absence of reward (i.e. explore new possibilities). This strategy captures the spirit of RL in moderating exploration versus exploitation without computing a continuous value function. Several indirect lines of evidence support this hypothesis (*Palidis et al., 2019*; *Izawa and Shadmehr, 2011*; *Shmuelof et al., 2012a*; *Shmuelof et al., 2012b*; *Dam et al., 2013*; *Wu et al., 2014*; *Nikooyan and Ahmed, 2015*; *Pekny et al., 2015*; *Vaswani et al., 2015*; *Cashaback et al., 2017*; *Cashaback et al., 2019*; *Chen et al., 2017*; *Dhawale et al., 2019*; *van der Kooij and Smeets, 2019*). For example, humans learn more efficiently if the structure of their natural movement variability aligns with the underlying learning objective (*Wu et al., 2014*), reaching movements become more variable during periods of low success rate (*Izawa and Shadmehr, 2011*; *Pekny et al., 2015*), and saccades become more variable in the absence of reward (*Takikawa et al., 2002*).

However, the relationship between variability and learning is nuanced. Although increasing motor variability may play a direct role in the induction of learning (*Dhawale et al., 2017*), one of the primary functions of motor learning is to reduce variability (*Crossman, 1959*; *Harris and Wolpert, 1998*; *Thoroughman and Shadmehr, 2000*; *Smith et al., 2006*; *Sternad and Abe, 2010*; *Verstynen and Sabes, 2011*). This two-sided relationship between learning and variability has made it challenging to tease apart behavioral and neural signatures of variability that degrade performance from those that improve it. A rigorous assessment of the interaction between learning and variability demands two important developments. First, we need a method for teasing apart sources of variability that hinder performance from those that facilitate learning. Second, we need to verify that the underlying neural circuits rely on reinforcement to regulate the variability along task-relevant dimensions. Here, we addressed these problems using a motor timing task in which monkeys produced different time intervals using different effectors. We first analyzed behavior and found evidence for multiple sources of motor variability, including memory drift and reward-dependent exploration. We then developed a generative model that could explain how reward regulates variability and facilitates learning. Finally, we probed the underlying neural circuits in multiple nodes of the cortico-basal ganglia circuits implicated in motor timing (*Wang et al., 2018*) and found that the variability across the population of thalamic neurons with projections to the dorsomedial frontal cortex (DMFC) was modulated by reward along task-relevant dimensions in a context-specific manner.

## Results

### Cue-Set-Go task

Two monkeys were trained to perform a Cue-Set-Go (CSG) motor timing task (*Figure 1A*). On each trial, animals had to produce either an 800 ms (Short) or a 1500 ms (Long) time interval ($t_t$) either with a saccade (Eye) or with a button press (Hand). The task thus consisted of four randomly interleaved trial types, Eye-Short (ES), Eye-Long (EL), Hand-Short (HS), and Hand-Long (HL). The trial type was cued at the beginning of each trial by a fixation cue ('Cue'). The Cue consisted of a circle and a square: red circle for ES, blue circle for EL, red square for HS, and blue square for HL (*Figure 1A*, right). After a random delay, a visual stimulus ('Tar') was flashed to the left or right of the screen. This stimulus specified the position of the saccadic target for the Eye trials and served no function in the Hand trials. After another random delay, the presentation of a 'Set' flash around the fixation spot indicated the start of the timing period. Animals had to proactively initiate a saccade or press a button such that the produced interval, $t_p$, between Set and movement initiation ('Go') would match $t_t$. Reward was provided when the relative error, defined as $e = (t_p - t_t)/t_t$ (*Figure 1A*) was within an experimentally-controlled acceptance window. On rewarded trials, the magnitude of reward decreased linearly with the size of the error. The width of the acceptance window was controlled independently for each trial type using a one-up-one-down staircase procedure (see Materials and methods) so that animals received reward on nearly half of the trials (*Figure 1C*, inset).

Animals learned to use the Cue and flexibly switched between the four trial types on a trial-by-trial basis (*Figure 1B*). There were no errors associated with using the wrong effector, and the number of errors with respect to the target interval was extremely small (~0.79% misclassified trials based on fits to a Gaussian mixture model). For both effectors, a robust feature of the behavior was that produced intervals ($t_p$) were more variable for the Long compared to the Short (*Figure 1C*). This is consistent with the common observation that timing variability scales with the interval being timed

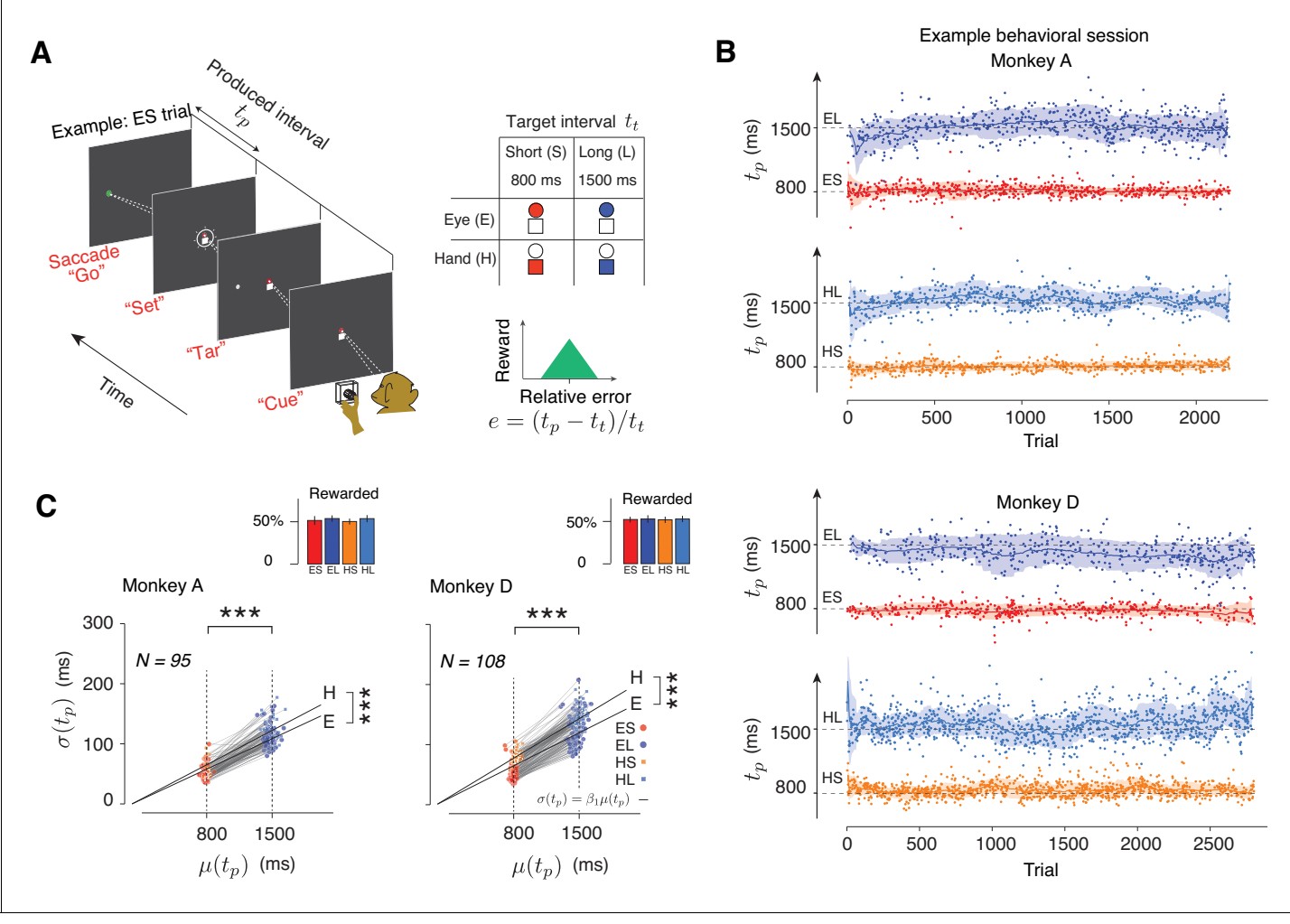

**Figure 1.** Task, behavior, and reward-dependency of variability. (A) The Cue-Set-Go task. (B) Behavior in a representative session for each of the two animals. For visual clarity, $t_p$ values (dots) for different trial types are shown along different abscissae for each trial type. The solid line and shaded area are the mean and standard deviation of $t_p$ calculated from a 50-trial sliding window. (C) The standard deviation of $t_p$ ($\sigma(t_p)$) as a function of its mean ($\mu(t_p)$) for each trial type in each behavioral session. Each pair of connected dots corresponds to Short and Long of the same effector in a single session. In both animals, the variability was significantly larger for the Long compared to the Short for both effectors (one-tailed paired-sample t test, ***p<<0.001, for monkey A, n = 190, $t_{128}$ = 157.4; for monkey D, n = 216, $t_{163}$ = 181.7). The solid black lines show the regression line relating $\sigma(t_p)$ to $\mu(t_p)$ across all behavioral sessions for each trial type ($\sigma(t_p) = \beta_1 \mu(t_p)$). Regression slopes were positive and significantly different from zero for both effectors ($\beta$ = 0.087 ± 0.02 mean±std for Eye and 0.096 ± 0.021 for Hand in Monkey A; $\beta_1$ = 0.10 ± 0.02 for Eye and 0.12 ± 0.021 for Hand in Monkey D). Hand trials were more variable than Eye ones (one-tailed paired-sample t-test, for monkey A, n = 95, $t_{52}$ = 6.92, ***p<<0.001, and for monkey D, n = 108, $t_{61}$ = 6.51, ***p<<0.001). Inset: The overall percentage of rewarded trials across sessions for each trial type.

(*Gibbon, 1977*; *Malapani and Fairhurst, 2002*). This *scalar variability* was evident in the linear relationship between mean ($\mu$) and standard deviation ($\sigma$) of $t_p$ in each behavioral session (*Figure 1C*, the slope of the linear test was significantly positive, one-tailed t-test, p<<0.001, for monkey A, df = 128, t = 32.12; for monkey D, p<<0.001, df = 163, t = 24.06).

## Deconstructing motor timing variability

The neurobiological basis of timing variability is not understood. Models of interval timing typically assume that timing variability is stationary and attribute it to various sources such as a noisy internal clock, a noisy accumulator, noisy oscillations, and noisy storage/retrieval mechanisms (*Gibbon et al., 1984*; *Killeen and Fetterman, 1988*; *Grossberg and Schmajuk, 1989*; *Church and Broadbent, 1990*; *Machado, 1997*; *Staddon and Higa, 1999*; *Jazayeri and Shadlen, 2010*; *Simen et al., 2011*; *Oprisan and Buhusi, 2014*). However, behavioral variability is typically nonstationary (*Weiss et al.,*

*1955*; *Merrill and Bennett, 1956*; *Gilden et al., 1995*), especially in the context of movements (*Chaisanguanthum et al., 2014*), reaction times (*Laming, 1979*), and interval timing (*Chen et al., 1997*; *Murakami et al., 2017*). These nonstationarities are particularly important as they can cause long-term drifts, unstable behavior, and poor performance. Therefore, it is important to tease apart the factors that underlie nonstationary variability, and understand how the brain maintains stable performance in spite of such variability.

We verified the presence of nonstationary behavior in our data by analyzing the serial correlations of $t_p$. In all four trial types, $t_p$ exhibited significant serial correlations up to a trial lag of 20 or more (p<0.01, dash lines: 1% and 99% confidence bounds by estimating the null distribution from shuffled series, *Figure 2A*) revealing strong slowly fluctuating sources of variability. Importantly, these correlations were stronger between trials with the same effector and same $t_t$ compared to trials that were

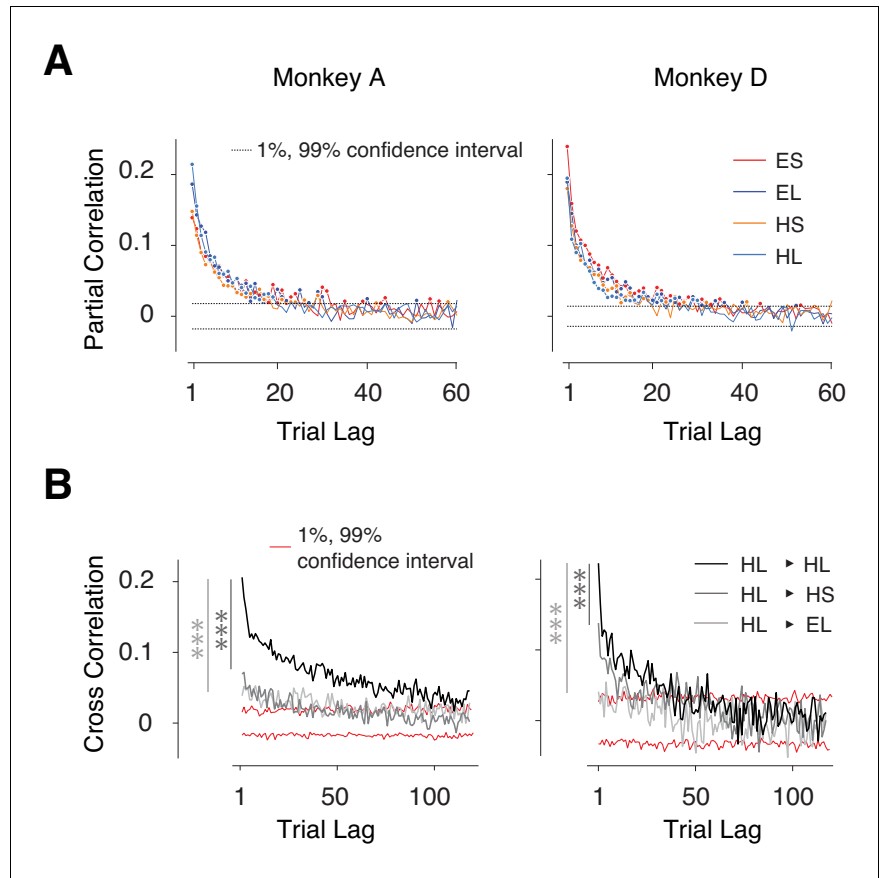

**Figure 2.** Context-dependent slow fluctuations of timing intervals. (**A**) Long-term correlation of $t_p$ across trials of the same type (same effector and target interval). For each behavioral session, and each trial type, we computed the partial correlation of $t_p$ as a function of trial lag. Each curve shows the average partial correlations across all behavioral sessions. Four types of trials are shown in different colors. Filled circles: correlation values that are larger than the 1% and 99% confidence interval (dashed line). (**B**) Examples showing the Pearson correlation coefficients of $t_p$'s as a function of trial lag. HL-HL indicates the correlation was averaged across trials transitioning between HL and HL; 1% and 99% confidence intervals were estimated from the null distribution. Serial correlations were stronger between trials with the same effector and interval compared to trials with the same effector but different interval (*** dark gray, paired sample *t*-test on two sets of cross-correlations with less than 20 trial lags and combining four trial types; monkey A: p<<0.001, n = 80, $t_{79}$ = 9.8; monkey D: p<<0.001, n = 80, $t_{79}$ = 5.8), and trials of the different effector but same interval (*** light gray, monkey A: p<<0.001, n = 80, $t_{79}$ = 6.7; monkey D: p<<0.001, n = 80, $t_{79}$ = 17.3). See *Figure 2—figure supplement 1A* for transitions between other conditions, and *Figure 2—figure supplement 1B* for a comparison of context-specificity with respect to saccade direction. The online version of this article includes the following figure supplement(s) for figure 2:

**Figure supplement 1.** Slow fluctuations of timing variability.

associated with different $t_t$ and/or different effectors (*Figure 2B*, *Figure 2—figure supplement 1A–B*). The context-specificity of these correlations suggests that they were not solely due to non-specific fluctuations of internal states such as the overall level of alertness, which should persist across the four randomly interleaved trial types.

Since performance in CSG depends on an accurate memory of $t_t$, we hypothesized that these fluctuations may be due to slow drifts in memory. To test this hypothesis, we reasoned that the fluctuations should be smaller if the demands on memory were reduced. Therefore, we trained two monkeys, not previously exposed to the CSG task, to perform a control task in which $t_t$ was measured on every trial, thereby minimizing the need to rely on memory (see Materials and methods). As expected, $t_p$ serial correlations diminished significantly in the control task (*Figure 2—figure supplement 1D*; p<<0.001, paired sample *t*-test, $t_{19}$ = 4.9, on the partial correlation between CSG and controls, trial lag was limited to 20 trials), suggesting that the slow fluctuations of $t_p$ in the CSG task were a signature of drift in memory. Note that our use of the term context (effectors and intervals) and memory (procedural) should not be confused with the notion of context (spatial) in classical studies of episodic memory. For the remainder of our analyses, we will refer to these fluctuations as memory drifts. Note that this drift reflects a random diffusion process with no explicit baseline and is consistent with broadening of the $t_p$ distribution.

## Reward regulates variability in a context-specific manner

Memory drifts, if left unchecked, could lead to large excursions away from $t_t$, and ruin performance. To maintain a reasonably accurate estimate of $t_t$, another process must counter the drift in memory. In CSG, the only cue that animals can rely on to maintain high performance is the trial outcome. Therefore, we hypothesized that animals rely on the trial outcome to counter memory drifts. In principle, the reward can calibrate behavior through a combination of directed and random explorations. In directed exploration, the agent repeats responses that previously led to better outcomes. In random explorations, the agent simply increases variability to explore responses that might yield better outcomes.

As a first step toward developing a model for how reward regulates behavior, we did a careful analysis of behavioral variability across trials. Specifically, we defined relative error on trial $n$ as $e^n = (t_p^n - t_t)/t_t$, and analyzed the relationship between error mean, $\mu(e^n)$, and error variability, $\sigma(e^n)$, as a function of error in the preceding trial ($e^{n-1}$). Results revealed that these error statistics were highly structured across trials: $\mu(e^n)$ increased monotonically with $e^{n-1}$, and $\sigma(e^n)$ had a U-shaped profile (*Figure 3B*, *Figure 3—figure supplement 1A*). We confirmed the statistical significance of these observations using linear regression for $\mu(e^n)$ and quadratic regression for $\sigma(e^n)$ (*Table 1*, see Materials and methods).

The monotonic relationship between $\mu(e^n)$ and $e^{n-1}$ is expected given the long-term correlations of $t_p$ caused by memory drifts. In contrast, the U-shaped relationship between $\sigma(e^n)$ and $e^{n-1}$ is unexpected and suggests that trial outcome regulates behavior variability. In particular, since smaller errors were associated with higher average rewards, this U-shaped relationship suggests that animals increase their behavioral variability when the reward is low and decrease it when the reward is high.

We performed a number of additional analyses to further scrutinize the unexpected relationship between $\sigma(e^n)$ and $e^{n-1}$. For example, we asked whether the modulation of $\sigma(e^n)$ could be due to differences across (as opposed to within) behavioral sessions. Applying the same analysis to different grouping of trials confirmed that the relationship between variability and reward was present within individual behavioral sessions (*Figure 3—figure supplement 1B*). Similarly, we asked whether the effect of reward on variability was binary (present/absent) or graded (larger variability for lower reward). A more refined binning of trials based on the magnitude of reward provided evidence that the effect was graded: $\sigma(e^n)$ was significantly correlated with the magnitude of the previous reward $r^{n-1}$ (Spearman's correlation, $\rho = -0.23$, ***p=2e-15, *Figure 3—figure supplement 1C*). Finally, the effect of reward was progressively weaker for larger trial lags (*Figure 3—figure supplement 1D–E*) revealing the lingering effects of reward (or lack thereof) beyond single trials. These results provide converging evidence in support of the relationship between reward and behavioral variability.

Our earlier analysis indicated that the slow memory drifts in the behavior were context-specific (*Figure 2B*, *Figure 2—figure supplement 1A–B*). For the reward modulation of variability to be effective, it must be also context-specific. That is, an increase in variability following a large error

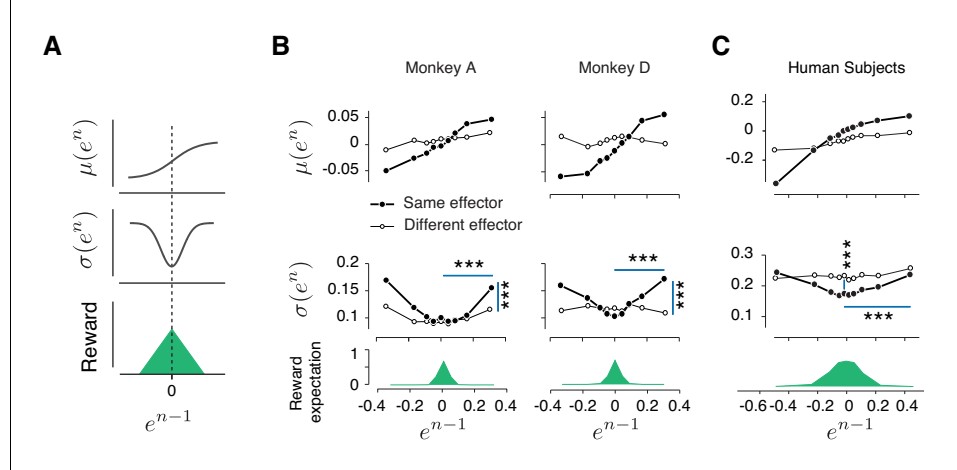

**Figure 3.** Rapid and context-dependent modulation of behavioral variability by reward. (**A**) Illustration of the expected effect of serial correlations and reward-dependent variability. Top: Positive serial correlations between produced intervals ($t_p$) creates a positive correlation between consecutive errors, and predicts a monotonic relationship between the mean of error in trial $n$, $\mu(e^n)$, and the value of error in trial $n-1$ ($e^{n-1}$). Middle: Variability decreases with reward. This predicts a U-shaped relationship between the standard deviation of $e^n$, $\sigma(e^n)$, and $e^{n-1}$. Bottom: Reward as a function of $e^{n-1}$. (**B**) Trial-by-trial changes in the statistics of relative error. Top: $\mu(e^n)$ as a function of $e^{n-1}$ in the same format shown in (**A**) top panel. Filled and open circles correspond to consecutive trials of the same and different types, respectively. Middle: $\sigma(e^n)$ as a function of $e^{n-1}$, sorted similarly to the top panel. Bottom: the reward expectation as a function of $e^{n-1}$. The reward expectation was computed by averaging the magnitude of reward received across trials. In the same effector, variability increased significantly after an unrewarded trials compared to a rewarded trials (horizontal line, two-sample F-test for equal variances, ***$p \ll 0.001$) for both large positive errors (Monkey A: $F(11169,10512) = 1.09$, Monkey D: $F(18540,13478) = 1.76$) and large negative errors (Monkey A: $F(8771,9944) = 1.40$, Monkey D: $F(21773,14889) = 1.62$). The variability after an unrewarded trial of the same effector was significantly larger than after an unrewarded trial of the other effector (vertical line, two-sample F-test for equal variances, ***$p \ll 0.001$) for both large positive errors (Monkey A: $F(11169,8670) = 1.20$, Monkey D: $F(18540,7969) = 1.32$) and large negative errors (Monkey A: $F(8771,5994) = 1.26$, Monkey D: $F(21773,7179) = 1.27$). (**C**) Same as (**B**) for human subjects. In humans, the variability was also significantly larger after a negatively reinforced trial compared to a positively reinforced trial (horizontal line, two-sample F-test for equal variances, ***$p \ll 0.001$) for both large positive errors ($F(5536,5805) = 1.19$) and large negative errors ($F(9366,9444) = 1.11$). The variability after a positively reinforced trial of the same effector was significantly lower than after a positively reinforced trial of the other effector (vertical line, two-sample F-test for equal variances, ***$p \ll 0.001$, $F(14497,15250) = 1.10$). For humans, the reward expectation was defined as the ratio between the number of trials with positive feedback and total number of trials.

The online version of this article includes the following figure supplement(s) for figure 3:

**Figure supplement 1.** The effect of reward on behavioral variability.

should only be present if the subsequent trial is of the same type. Otherwise, reinforcement of one trial type, say ES, may incorrectly adjust variability in the following trials of another type, say HL, and would interfere with the logic of harnessing reward to calibrate the memory of $t_t$. To test this possibility, we analyzed error statistics between pairs of consecutive trials associated with different types

**Table 1.** Quantitative assessment of the dependence of $\mu(e^n)$ and $\sigma(e^n)$ on $e^{n-1}$.

| | Monkey A | Monkey D | Humans |
|---|---|---|---|
| **Parameters** | **Same vs. different effector** | **Same vs. different effector** | **Same vs. different effector** |
| $m_1$ | 0.16 [0.12, 0.19]>0.04 [0.03, 0.06] | 0.20 [0.12, 0.28]>−0.00 [-0.05, 0.04] | 0.35 [0.27, 0.44]>0.15 [0.09, 0.20] |
| $m_0$ | 0.00 [-0.00, 0.01]~−0.01 [0.01, 0.01] | −0.00 [-0.02, 0.01]~−0.01 [0.00, 0.02] | −0.03 [-0.08, 0.02]~−0.06 [-0.08,−0.05] |
| $s_2$ | 0.50 [0.39, 0.60]>0.24 [0.14, 0.33] | 0.48 [0.19, 0.78]>−0.06 [-0.17, 0.05] | 0.31 [0.21, 0.42]>0.06 [-0.03, 0.16] |
| $s_1$ | −0.02 [-0.04, 0.01]~0.02 [-0.00, 0.04] | 0.03 [-0.03, 0.10]~−0.01 [-0.04, 0.01] | 0.00 [-0.03, 0.04]~0.03 [-0.00, 0.06] |
| $s_0$ | 0.09 [0.09, 0.10]~0.08 [0.08, 0.92] | 0.11 [0.10, 0.12]~0.12 [0.11, 0.12] | 0.17 [0.16, 0.19]<0.22 [0.22, 0.24] |

(*Figure 3B* open circles; *Table 1*). Results indicated that (1) correlations were strongest between pairs of trials of the same type as expected from our previous cross-correlation analysis in (*Figure 2B*) and (2) the modulation of variability was most strongly and consistently present across trials of the same type. Together, these results provide strong evidence that animals increase behavioral variability in the next trial in a context-dependent manner to systematically promote exploration.

## Reward-dependent context-specific regulation of variability in humans

To ensure that our conclusions were not limited to data collected from highly trained monkeys, we performed a similar experiment in human subjects. In human psychophysical experiment, $t_t$ varied from session to session, and subjects had to constantly adjust their $t_p$ by trial-and-error. Similar to monkeys, human behavior exhibited long-term serial correlations and these correlations were context (effector) specific (*Figure 2—figure supplement 1B–C*). We performed the same analysis as in monkeys to characterize the dependence of $\mu(e^n)$ and $\sigma(e^n)$ on $e^{n-1}$. Results were qualitatively similar: $\mu(e^n)$ increased monotonically with $e^{n-1}$ verifying the presence of slow drifts, and $\sigma(e^n)$ had a U-shaped with respect to $e^{n-1}$ indicating that subjects used the feedback to regulate their variability (*Figure 3C* and *Table 1*). Finally, similar to monkeys, the effect of reward on variability was context-specific (*Figure 3C*). This result suggests that the memory of a time interval is subject to slow drifts and humans and monkeys use reward-dependent regulation of variability as a general strategy to counter these drifts and improve performance.

We used linear regression ($\mu(e^n) = m_0+m_1 e^{n-1}$) to relate $\mu(e^n)$ to $e^{n-1}$, and quadratic regression ($\sigma(e^n) = s_0+s_1 e^{n-1}+s_2(e^{n-1})^2$) to relate $\sigma(e^n)$ to $e^{n-1}$. Fit parameters and the corresponding confidence intervals [1%, 99%] are tabulated for each monkey and for humans. We compared the magnitude of fit parameters between the same versus different effector conditions (bold: significantly different).

## Trial outcome causally impacts behavioral variability

So far, our results establish a relationship between trial outcome and behavioral variability. However, since in our experiment error size and trial outcome were deterministically related (larger error led to reduced reward), it is not possible to determine which of these two factors had a causal impact on behavioral variability. More specifically, our current findings are consistent with two interpretations. In one model, trial outcome regulates behavioral variability (*Figure 4A*; 'Causal'). In another, variability is influenced by various upstream factors such as error magnitude, but not by trial outcome (*Figure 4A*; 'Correlational'). To distinguish between these two models, we carried out a new human psychophysics experiment (*Figure 4B*) in which trial outcome was probabilistic (*Figure 4C*) so that the presence or absence of a 'correct' feedback was not fully determined by the magnitude of the error. This design enabled us to analyze the effect of trial outcome ('correct' versus 'incorrect') on behavioral variability .

We compared behavioral variability after 'correct' and 'incorrect' trials separately for small (*Figure 4D*, left) and large (*Figure 4D*, right) errors. Across subjects and irrespective of the size of error, variability was significantly larger after incorrect compared to correct trials (*Figure 4D*). In other words, the presence or absence of reward had a causal impact on behavioral variability that was not mediated through the size of error. This result substantiates our hypothesis and provides evidence against all models that do not account for the direct dependence of behavioral variability on the trial outcome.

## A generative model linking multiple timescales of variability to reward-based learning

Our results so far indicate that errors in $t_p$ are governed by two key factors: long-term serial correlations creating local and persistent biases in $t_p$, and short-term modulations of $t_p$ variability by reward. Intuitively, this could provide the means for an efficient control strategy: when bias due to memory drift increases, error increases, reward drops, and the animal seeks to find the correct target interval by increasing variability. Indeed, several recent behavioral experiments have found evidence that is qualitatively consistent with this control strategy (*Izawa and Shadmehr, 2011*; *Shmuelof et al., 2012a*; *Dam et al., 2013*; *Wu et al., 2014*; *Nikooyan and Ahmed, 2015*; *Pekny et al., 2015*; *Chen et al., 2017*; *Cashaback et al., 2019*).

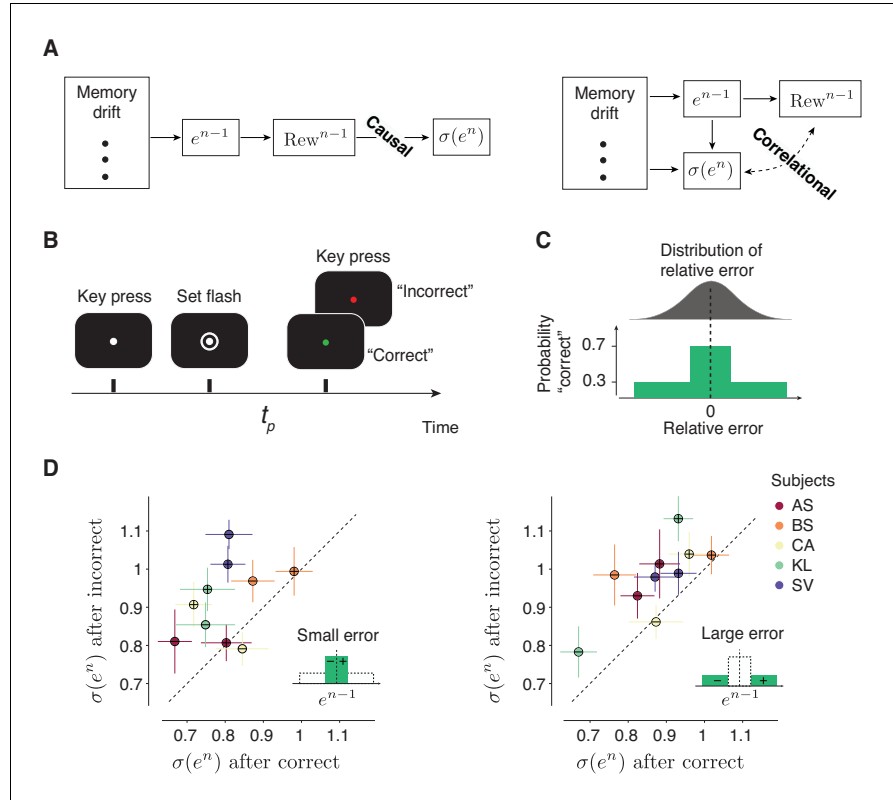

**Figure 4.** Causal effect of reward on behavioral variability. (A) Alternative interpretations of the relationship between the outcome of the preceding trial ($Rew^{n-1}$) and behavioral variability in the current trial denoted $\sigma(e^n)$. Left: A model in which $Rew^{n-1}$ causally controls $\sigma(e^n)$. In this model, various factors (e.g.,memory drift) may determine the size of error ($e^{n-1}$), $e^{n-1}$ determines $Rew^{n-1}$, and $Rew^{n-1}$ directly regulates $\sigma(e^n)$. Right: A model in which the relationship between $Rew^{n-1}$ and $\sigma(e^n)$ is correlational. In this model, $Rew^{n-1}$ is determined by $e^{n-1}$, and $\sigma(e^n)$ may be controlled by various factors (including $e^{n-1}$) but not directly by $Rew^{n-1}$. (B,C) A time interval production task with probabilistic feedback to distinguish between the two models in (A). (B) Trial structure. The subject has to press the spacebar to initiate the trial. During the trial, the subject is asked to hold their gaze on a white fixation spot presented at the center of the screen. After a random delay, a visual ring ('Set') is flashed briefly around the fixation spot. The subject has to produce a time interval after Set using a delayed key press. After the keypress, the color of the fixation spot changes to red or green to provide the subject with feedback (green: 'correct', red: 'incorrect'). (C) Top: A schematic illustration of the distribution of relative error , computed as ($t_p$–$t_t$)/$t_t$. Bottom: After each trial, the feedback is determined probabilistically: the subject is given a 'correct' feedback with the probability of 0.7 when $t_p$ is within a window around $t_t$, and with the probability of 0.3 when errors are outside this window. The window length was adjusted based on the overall behavioral variability so that each subject receives approximately 50% 'correct' feedback in each behavioral session. (D) The causal effect of the outcome of the preceding trial on behavioral variability in the current trial. Left: Scatter plot shows the behavioral variability after 'incorrect' (ordinate) versus 'correct' (abscissa) trials, for all five subjects, after trials in which $e^{n-1}$ was small (inset). Results for the positive and negative errors are shown separately (with '+' and '-' symbols, respectively). Right: Same as the left panel for trials in which $e^{n-1}$ was large (inset). In both panels, the variability across subjects was significantly larger following incorrect compared to correct trials (p<<0.001, paired sample $t$-test, $t_{199}$ = 12.8 for small error, and $t_{199}$ = 13.7 for large error, see Materials and methods).

To assess this idea rigorously, we aimed to develop a generative model that could emulate this control strategy. We reasoned that the generative process must have two distinct components, one associated with long-term serial correlations due to the memory drift, and another, with the short-term effect of reward on variability. Accordingly, we sought to develop a Gaussian process (GP) model that could capture both effects. This choice was motivated by three factors: (1) GPs automatically describe observations up to their second order statistics, which are the relevant statistics in our data, (2) GPs offer a nonparametric Bayesian fit to long-term serial correlations, and (3) as we will describe, GPs can be readily augmented to regulate variability based on reward.

GPs are characterized by a covariance matrix – also known as the GP kernel – that specifies the degree of dependence between samples, and thus determines how slowly the samples fluctuate. The most common and mathematically convenient formulation of this kernel is known as the 'squared exponential' kernel function, denoted $K_{SE}$ (*Figure 5A*, top left):

$$K_{SE}(n, n-r) = \exp(-\frac{r^2}{2l_{SE}^2})$$

In $K_{SE}$, the covariance between any two samples (indexed by $n$ and $n-r$) drops exponentially as a function of temporal distance ($r$) and the rate of drop is specified by the *characteristic length parameter*, $l_{SE}$. When $l_{SE}$ is small, samples are relatively independent, and when it is large, samples exhibit long-range serial correlations (*Figure 5A*, left, second and third rows).

Using GPs as the backbone of our model, we developed a reward-sensitive GP (RSGP) whose kernel ($K_{RSGP}$) is the weighted sum of two kernels, a classic squared exponential kernel ($K_{SE}$) scaled by

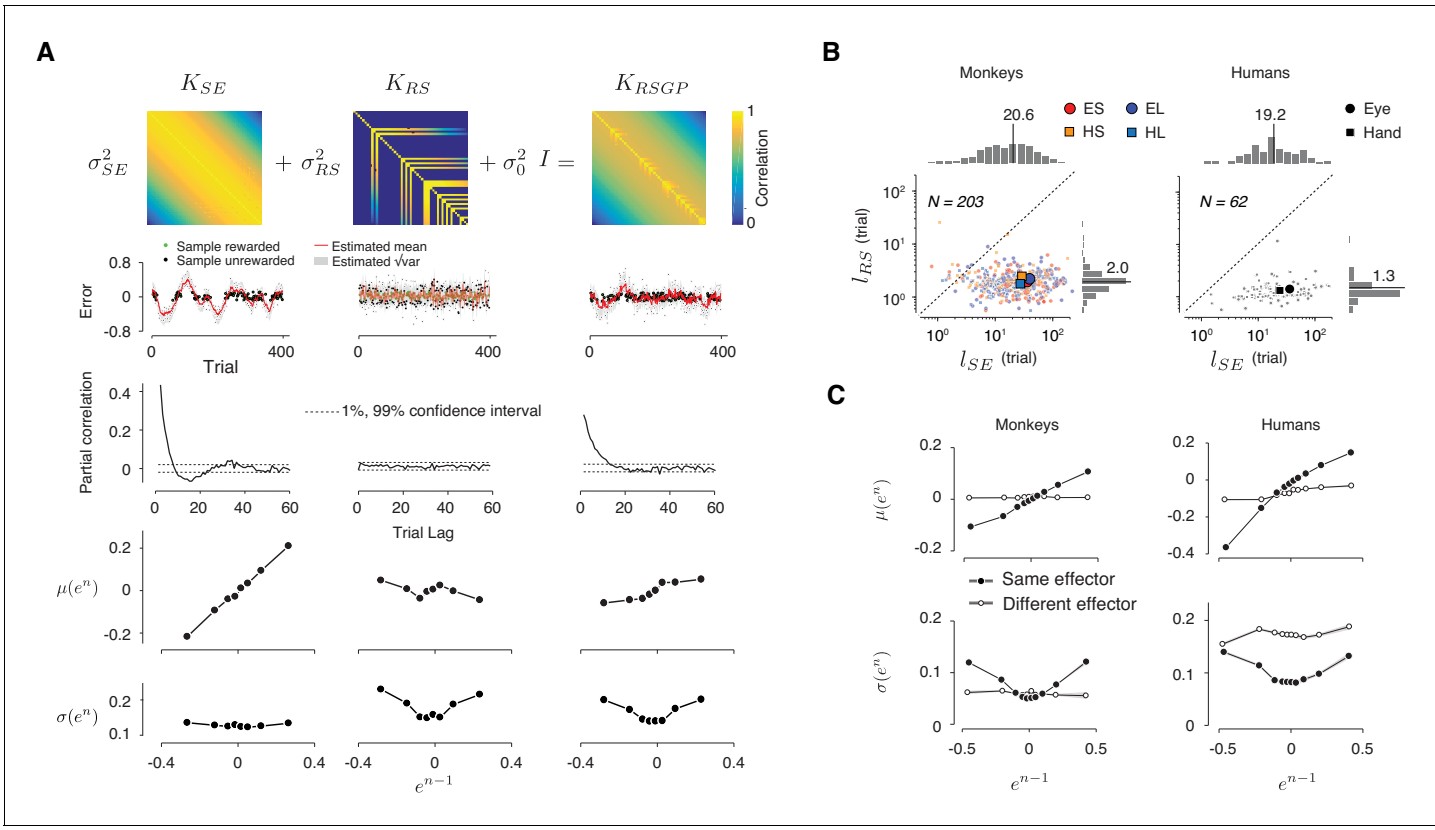

**Figure 5.** A reward-sensitive Gaussian process model (RSGP) capturing reward-dependent control of variability. (**A**) Top: The covariance function for the RSGP model ($K_{RSGP}$) is the sum of a squared exponential kernel ($K_{SE}$), a reward-dependent squared exponential kernel ($K_{RS}$) and an identity matrix (*I*) weighted by $\sigma_{SE}^2$, $\sigma_{RS}^2$, and $\sigma_0^2$, respectively. Second row: Simulations of three GP models, one using $K_{SE}$ only (left), one using $K_{RS}$ only (middle), and one with the full $K_{RSGP}$ (right). Third row: Partial correlation of samples from the three GPs in the second row. Fourth and fifth rows: The relationship between the mean (fourth row) and standard deviation (fifth row) of $e^n$ as a function of $e^{n-1}$ in the previous trial, shown in the same format as in *Figure 3B*. Only the model with full covariance function captures the observed behavioral characteristics. (**B**) Length scales, $l_{SE}$ and $l_{RS}$ associated with $K_{ES}$ and $K_{RS}$, respectively, derived from fits of RSGP to behavioral data from monkeys (left) and humans (right). Small and large symbols correspond to individual sessions and the median across sessions, respectively. Different trial types are shown with different colors (same color convention as in *Figure 1B*). $l_{RS}$ was significantly smaller than the $l_{SE}$ (monkeys: p<<0.001, one-way ANOVA, $F_{1, 945} = 463.4$; humans: p<<0.001, one-way ANOVA, $F_{1, 235} = 102.5$). (**C**) Statistics of the predicted behavior from the RSGP model fits, shown in the same format as *Figure 3B,C*. Data were generated from forward prediction of the RSGP model fitted to behavior (see Materials and methods for details). The standard error of the mean computed from n = 100 repeats of sampling of trials is shown as a shaded area, but it is small and difficult to discern visually.

The online version of this article includes the following figure supplement(s) for figure 5:

**Figure supplement 1.** RSGP model fits.

**Figure supplement 2.** Analysis of error statistics in the probabilistic reward experiment.

$\sigma^2_{SE}$, and a reward-sensitive kernel ($K_{RS}$) scaled by $\sigma^2_{RS}$ (*Figure 5A*, top middle). A third diagonal matrix ($\sigma^2_0 I$) was also added to adjust for baseline variance:

$$K_{RSGP}(n, n-r) = \sigma^2_{SE} K_{SE}(n, n-r) + \sigma^2_{RS} K_{RS}(n, n-r) + \sigma^2_0 I$$

$K_{RS}$ also has a squared exponential form with a length parameter, $l_{RS}$. However, the covariance terms were non-zero only for rewarded trials (*Figure 5A*, top middle). The reward was a binary variable determined by an acceptance window around $t_t$. This allows rewarded samples to have more leverage on future samples (i.e. larger covariance), and this effect drops exponentially for trials farther in the future.

Intuitively, RSGP operates as follows: $K_{SE}$ captures the long-term covariance across samples (*Figure 5A*, 3rd row, left). $K_{RS}$ regulates shorter-term covariances (*Figure 5A*, 3rd row, middle) and allows samples to covary more strongly with recent rewarded trials (*Figure 5A*, bottom, middle). Non-zero values in $K_{RS}$ after rewarded trials strengthen the correlation between samples and lead to a reduction of variability, whereas zero terms after unrewarded trials reduce correlations and increase variability. Using simulations of the full model with $K_{RSGP}$ as well as reduced models with only $K_{SE}$ or $K_{RS}$ (*Figure 5A*, second row), we verified that both kernels were necessary and that the full RSGP model was capable of capturing the two key features (*Figure 5A*, fourth and fifth rows). Moreover, we used simulations to verify that parameters of the model were identifiable, that is, fits of the model parameters to simulated data robustly recovered the ground truth (*Table 2*).

We fitted the RSGP model to the behavior of both monkeys and humans (*Figure 5B*, *Figure 5—figure supplement 1*). The recovered characteristic length associated with serial correlations ($l_{SE}$) were invariably larger than that of the reward-dependent kernel ($l_{RS}$) (Monkeys: $l_{SE} = 20.6 \pm 21.4$, $l_{RS} = 2.0 \pm 0.7$; Humans: $l_{SE} = 19.2 \pm 21.8$, $l_{RS} = 1.3 \pm 0.4$; Median ±MAD). The model fit of variances ($\sigma_{SE}$, $\sigma_{RS}$ and $\sigma_0$) are shown in *Figure 5—figure supplement 1C*. In monkeys, $\sigma_0$ and $\sigma_{SE}$ but not $\sigma_{RS}$ were significantly different between two effectors ($\sigma_0$: p<<0.001, one-tail two sample $t$-test, df = 482, t = 5.26; $\sigma_{SE}$: p<<0.001, two sample $t$-test, df = 482, t = 5.06; $\sigma_{RS}$: p=0.13, $t$-test, df = 261, t = 1.5 for the Short interval, and p=0.26, $t$-test, dt = 219, t = 1.1 for the Long interval). The dependence of $\sigma_0$ and $\sigma_{SE}$ on effector was consistent with our session-wide analysis of variance (*Figure 1C*). In the human subjects, variance terms were not significantly different between effectors (p=0.03 for $\sigma_0$, p=0.01 for $\sigma_{SE}$, and p=0.027 for $\sigma_{RS}$, two sample $t$-test, df = 118). Importantly, across both monkeys and humans, the model was able to accurately capture the relationship of $\mu(e^n)$ and $\sigma(e^n)$ to $e^{n-1}$ (*Figure 5C*, and Materials and methods). These results validate the RSGP as a candidate model for simultaneously capturing the slow fluctuations of $t_p$ and the effect of reward on $t_p$ variability.

Two aspects of the RSGP model are noteworthy. First, $K_{RS}$ was formulated to capture the effect of reward between trials of the same type, and not transitions between trials associated with different types. The addition of kernels for transitions between effectors was unnecessary because the effect of reward was effector-specific (*Figure 3B*, *Figure 3—figure supplement 1A*). Second, the RSGP was also able to capture the modulation of behavioral variability in the probabilistic reward experiment (*Figure 5—figure supplement 2B*).

## Alternative models

To validate our interpretation of behavioral data in terms of the RSGP model, we tested alternative models that might account for the three salient features of the data: (1) long-range $t_p$ correlations, (2) the monotonic relationship between $\mu(e^n)$ and $e^{n-1}$, and (3) the U-shaped relationship between $\sigma(e^n)$ and $e^{n-1}$. The first class of alternative models tested the hypothesis that modulations of $\sigma(e^n)$ were controlled by the magnitude of preceding errors, and not by the corresponding rewards

**Table 2.** Ground truth versus model fits for the hyperparameters used in RSGP simulation in *Figure 5A* and *Figure 5—figure supplement 1A*.

|  | $l_{SE}$ | $\sigma_{SE}$ | $l_{RS}$ | $\sigma_{RS}$ | $\sigma_0$ |
|---|---|---|---|---|---|
| Ground truth | 20.0 | 0.141 | 2.0 | 0.141 | 0.10 |
| MML fit | 18.4 | 0.129 | 2.14 | 0.137 | 0.0749 |

(*Figure 4A* Right). In this class, we considered several types of GP models without explicit sensitivity to reward, autoregressive models that exhibit serial correlations with no reward-dependency (*Wagenmakers et al., 2004*), and models whose behavior was governed by a mixture of stochastic processes (e.g. Gaussian process mixture models with different variances). None of these models were able to capture all the aforementioned features but we will not discuss these results further since our probabilistic reward control experiment already substantiated the causal role of reward on $\sigma(e^n)$ (*Figure 4D*).

Next, we considered models that took the reward into account explicitly (*Kaelbling et al., 1996*; *Sutton and Barto, 1998*) but whose updating algorithms differed from the RSGP model (*Figure 6*). RL models are usually geared toward problems with discrete action spaces such as multi-armed bandit problems (*Dayan and Daw, 2008*). Here, we adapted two such RL-based models to our task and tested whether they could capture the behavioral characteristics of humans (*Figure 6A*) and monkeys (*Figure 6B*).

## RL-based sampling (MCMC model)

The sampling model, which was proposed by *Haith and Krakauer, 2014* is an adaptation of the epsilon-greedy strategy to a continuous variable. In our implementation, the model uses previous trials to estimate the most rewarding target interval, denoted $t_t{}^*$, and generates a new target interval for the current trial by sampling from a Gaussian distribution centered at $t_t{}^*$. The sample is further subjected to scalar noise to produce $t_p$, and the outcome of the current trial is used to update $t_t{}^*$.

Using simulations, we verified that this model allows the agent to maintain $t_t{}^*$ close to the actual target interval, $t_t$, despite the stochasticity introduced by sampling and the production noise due to scalar variability (*Figure 6C*, *Figure 6—figure supplement 1A*). However, when the model was parameterized to capture the monotonic relationship between $\mu(e^n)$ and $e^{n-1}$, it could not capture the U-shaped relationship between $\sigma(e^n)$ and $e^{n-1}$ ($s_2$ = 0.045 [-0.048 0.14], *Figure 6C*, middle).

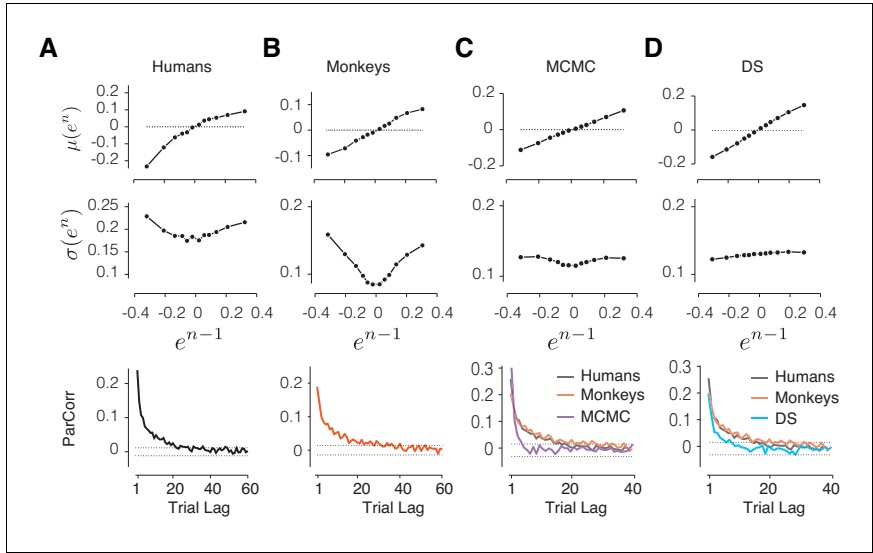

**Figure 6.** Comparison of alternative models to behavioral data and the RSGP model. (**A**) Human behavior. Top: Same results as *Figure 3* showing the monotonic relationship between the mean of error in trial $n$, $\mu(e^n)$, and the value of error in trial $n$-$1$ ($e^{n-1}$). Middle: Same as *Figure 3* showing the U-shaped relationship between the standard deviation of $e^n$, $\sigma(e^n)$, and $e^{n-1}$. Bottom: Same as *Figure 2A* showing partial correlation between produced intervals. (**B**) Same as (**A**) for data pooled across the animals. (**C**) Same as (**A**) for simulations of the RL-based Markov chain Monte Carlo (MCMC) sampling model. The data for humans and monkeys are included in the bottom panel for comparison. (**D**) Same as (**C**) for the RL-based directed search (DS) model based on the subset of simulation that did not exhibit run-away behavior.

The online version of this article includes the following figure supplement(s) for figure 6:

**Figure supplement 1.** Example RL-based model behavior.

Based on these discrepancies, we concluded that the RL-based sampling is not consistent with how humans and monkeys regulate their timing behavior.

## RL-based directed search (DS model)

This model is a variant of RL in which the agent relies on the reward gradient to perform directed exploration. To adapt this model to our task, we built a generative process that adjusts $t_t*$ based on the reward gradient ($\Delta r = r^{n-1} - r^{n-2}$) and interval gradient ($\Delta t_t = t_t^{n-1}* - t_t^{n-2}*$) in the two preceding trials using the following algorithm: $t_t^{n}* = t_t^{n-1}* + \alpha \cdot \Delta r \cdot \Delta t_t + n_e$, where $\alpha$ represents the learning rate and $n_e$ is the estimation noise (see Materials and methods). This algorithm moves $t_t^{n}*$ in the same direction as the previous trial when $\Delta r$ is positive (reward-increase), reverses direction when $\Delta r$ is negative (reward-decrease), and does not change $t_t^{n}*$ when there is no change in reward. Finally, $t_p^{n}$ is generated with a scalar production noise.

We simulated the DS model using a wide range of parameters for both the learning rate and noise. Overall, the model was able to adjust behavior based on reward (*Figure 6—figure supplement 1B*). However, upon close examination, we found the behavior of the DS model to be unstable. Specifically, unlike the MCMC and RSGP models (*Figure 6—figure supplement 1A,C*), the DS model could easily lose track of the target interval and exhibit 'run-away' behavior, that is, generate $t_p$ values that deviated from $t_t$ for an extended number of consecutive trials (*Figure 6—figure supplement 1B*). Simulating the model with the parameters fit to animals' behavior, the probability of run-away behavior in a typical 2000 trial session was $65.1 \pm 1.2\%$ (see Materials and methods). This brittleness occurs because the updating rule in the DS model cannot implement the directed search when multiple consecutive trials are unrewarded.

To further assess the DS model, we focused on the subset of DS simulations that did not exhibit run-away behavior, and asked whether they were able to reproduce the key features in our data. For this subset of DS simulations, the model was able to capture the dependencies between consecutive trials relating $\mu(e^n)$ to $e^{n-1}$ (*Figure 6D*; $m_1 = 0.46$ [0.44 0.48]), but not the U-shaped relationship between $\sigma(e^n)$ and $e^{n-1}$ (*Figure 6D*; $s_2 = -0.046$ [$-0.061$ $-0.03$]).

We also note that both the MCMC and DS models were unable to capture the long-term correlations that were present in the behavioral data and the simulations of the RSGP model (*Figure 6*, bottom). This shortcoming can be potentially rectified by adding a slow process to the MCMC and DS models. However, this modification would not be able to rescue the inability of these alternative models in capturing the U-shaped relationship between $\sigma(e^n)$ and $e^{n-1}$.

## Directed versus random exploration

Since RSGP was able to capture the full gamut of behavioral characteristics associated with the animals' exploratory behavior, we asked whether these explorations were directed or random. To do so, we performed extensive analyses comparing the behavior of the RSGP model to that of the MCMC and DS models. MCMC and DS can be viewed as autoregressive models whose coefficients depend on past rewards. The MCMC sets all coefficients to zero except a single trial in the past returned by the Metropolis-Hastings sampling algorithm. The DS sets all coefficients to zero except the last two that are determined based on the corresponding reward gradient and a fixed learning rate. The RSGP can also be written in terms of an autoregressive model with nonstationary reward-dependent coefficients and noise. However, the key feature that distinguishes the RSGP is that it performs Bayesian inference over $t_t$. The covariance function (or kernel) of the Gaussian process defines the coefficients and acts as a prior for future samples. The reward-sensitive kernel in the RSGP allows the coefficients to be updated continuously based on the history of past trial outcomes. When the reward rate is high, RSGP implements a strategy that is akin to directed exploration: it increases its reliance on the prior and drives responses toward previously rewarded trials. In contrast, when the reward rate is low, the RSGP relies more on random explorations: it generates samples from a wider distribution. Therefore, RSGP strikes an optimal balance (in the Bayesian sense) between bias (directed exploration) and variance (random exploration) as needed by the history of outcomes.

The interpretation of the effect of reward in terms of both directed and random explorations predicts that the monotonic relationship between $\mu(e^n)$ and $e^{n-1}$, which we originally explained in terms of memory drift, should additionally be sensitive to trial outcome. To test this prediction, we

returned to our probabilistic reward experiment in which trial outcome was dissociated from error magnitude, and asked whether the relationship between $\mu(e^n)$ and $e^{n-1}$ was modulated by feedback. Remarkably, results confirmed this prediction. The monotonic relationship between $\mu(e^n)$ and $e^{n-1}$ was stronger after trials that subjects received positive feedback (*Figure 5—figure supplement 2A*), and this effect was consistent with the behavior of the RSGP model (*Figure 5—figure supplement 2B*). These results indicate that humans and monkeys use a combination of directed and random exploration strategies in accordance with the RSGP model that uses past rewards to adjust both the mean and variance of future responses.

## Slow fluctuations in the cortico-basal ganglia circuits

Recently, we identified a cortico-basal ganglia circuit that plays a causal role in animals' performance during a flexible timing behavior (*Wang et al., 2018*). A combination of population data analysis and modeling revealed that animals control the movement initiation time by adjusting a tonic signal in the thalamus that sets the speed at which activity in the dorsomedial frontal cortex (DMFC) and caudate evolve toward an action-triggering state. Based on these results, we hypothesized that memory drifts in behavior may be accompanied by drifts of neural activity in these areas.

To test this hypothesis, we recorded separately from populations of neurons in candidate regions of DMFC, caudate and thalamus. Based on previous work suggesting the importance of initial conditions in movement initiation time (*Carpenter and Williams, 1995*; *Churchland et al., 2006*; *Jazayeri and Shadlen, 2015*; *Hauser et al., 2018*; *Lara et al., 2018*; *Remington et al., 2018b*), we asked whether neural activity near the time of Set (i.e. onset of motor timing epoch) could be used to predict the slow fluctuations in error, which we denote by $e_{slow}$ (*Figure 7A*, Right, red line). To estimate $e_{slow}$ on a trial-by-trial basis, we relied on the RSGP model, which readily decomposed the $t_p$ time series into a slow memory-dependent and a fast reward-dependent component. We fitted the RSGP in a context-specific manner and then inferred the value of $e_{slow}$ for each trial in each session (see Materials and methods).

We measured spike counts within a 250 ms window before Set (*Figure 7A*, Left), denoted $r$, and formulated a multi-dimensional linear regression model to examine the trial-by-trial relationship between $r$ and $e_{slow}$. We computed the regression weight, $\beta$, that when multiplied by $r$, would provide the best linear fit to $e_{slow}$. Considering the vector of spike counts in each trial as a point in a coordinate system where each axis corresponds to the activity of one neuron ('state space'), $\beta$ can be viewed as the direction along which modulations of spike counts most strongly correlate with memory drifts. Accordingly, we will refer to the direction associated with $\beta$ as the drift direction and will denote the strength of activity along that direction by $z$ ($z = r\beta$, *Figure 7A*, Right, blue line).

To test the hypothesis that memory drifts were accompanied by drifts of neural activity in the thalamus, DMFC, and caudate, we used a cross-validated procedure (see Materials and methods) to compute $\beta$, derive a trial-by-trial estimate of $z$, and measure the correlation between $z$ and $e_{slow}$ on a session-by-session basis. We found strong evidence that neural activity in all three areas exhibited slow fluctuations in register with memory drifts. In 91%, ~79%, and ~45% of sessions, the activity along the drift direction in the thalamus ($z_{Th}$), DMFC ($z_{DMFC}$), and caudate ($z_{Cd}$), respectively, was significantly correlated with $e_{slow}$ (*Figure 7B–D*, p<0.01, null hypothesis test by shuffling trials).

## Reward-dependent regulation of variability in the thalamus

Next, we analyzed the statistics of neurally inferred drift ($z$) across pairs of consecutive trials using the same approach we applied to error statistics in behavioral data (*Figure 3B*). To do so, we extracted pairs of ($e^{n-1}$, $z^n$) for consecutive trials of the same type and binned them depending on the value of $e^{n-1}$. Because all three areas carried a signal reflecting memory drifts, we expected $\mu(z^n)$ in all the areas to have a monotonic relationship with $e^{n-1}$. Results were consistent with this prediction as evidenced by the slope of a linear regression model relating $\mu(z^n)$ to $e^{n-1}$ (filled circles in *Figure 8B–D* top, *Table 3*). Moreover, this relationship was absent for consecutive trials associated with different effectors (open circles in *Figure 8B–D* top, *Table 3*). These results substantiate the presence of context-specific fluctuations of neural activity in register with drifts in animals' memory of $t_t$ across populations of neurons in the thalamus, DMFC and caudate.

A critical question was whether, in any of these areas, reward regulates the variability of neural activity. Importantly, the reward-dependent modulation of variability should be restricted to the drift

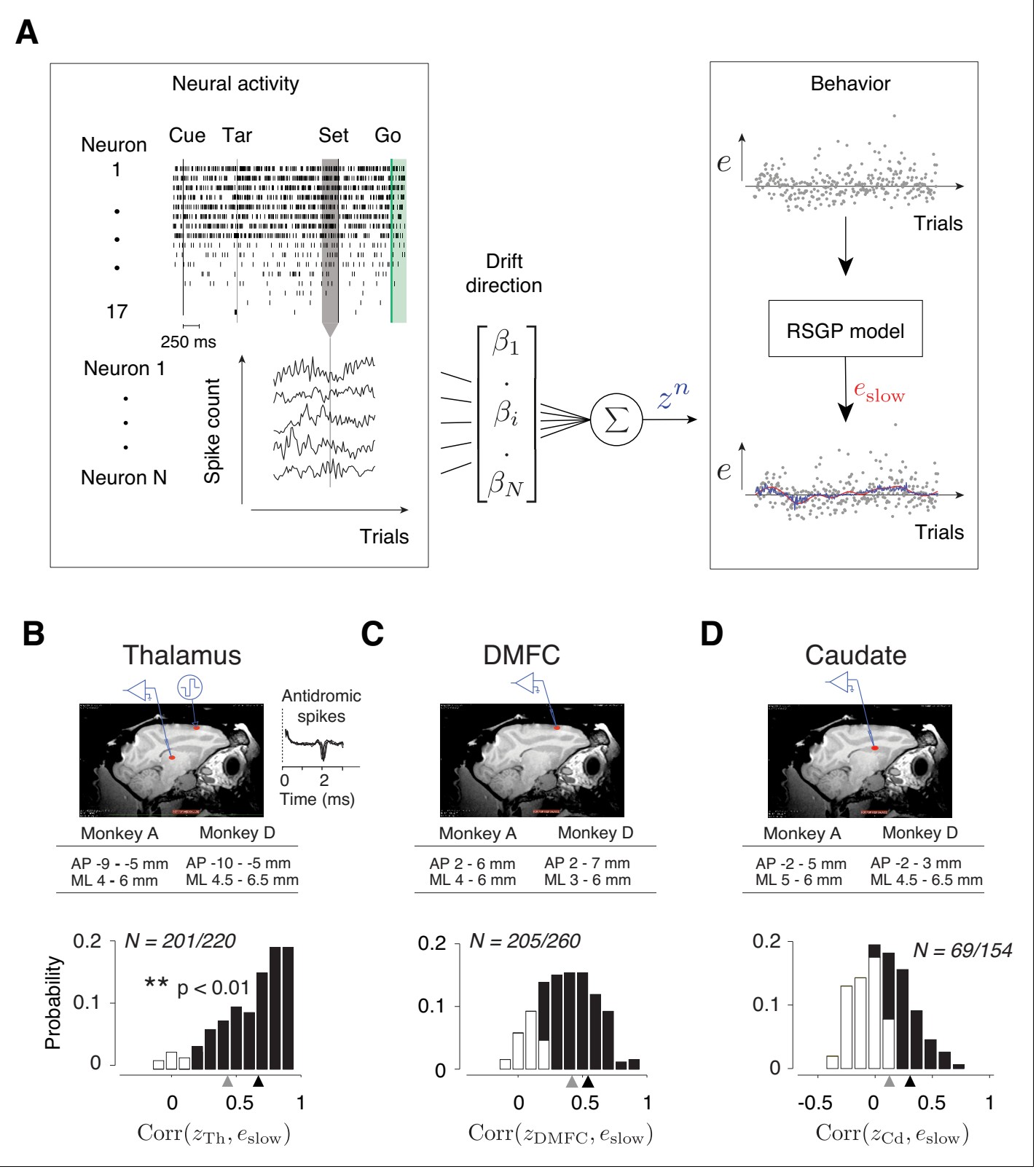

**Figure 7.** Representation of slow fluctuations of behavior in population activity. (**A**) The schematics of the analysis for identifying the drift direction across a population of simultaneously recorded neurons. Top left: The rows show spike times (ticks) of 17 simultaneously recorded thalamic neurons in an example trial. From each trial, we measured spike counts within a 250 ms window before Set (gray region). Bottom left: The vector of spike counts from each trial (gray vertical line) was combined providing a matrix containing the spike counts of all neurons across all trials. Middle: The spike counts

*Figure 7 continued on next page*

*Figure 7 continued*

across trials and neurons were used as the regressor in a multi-dimensional linear regression model with weight vector, $\boldsymbol{\beta}$, to predict the slow component of error ($e_{slow}$). Right: We fitted the RSGP to errors (black dots, $e$) and then inferred $e_{slow}$. The plot shows the neural prediction ($z^n$, blue) overlaid on $e_{slow}$ derived from RSGP fit to behavior (red line). (B) Top: Parasagittal view of one of the animals (monkey D) with a red ellipse showing the regions targeted for electrophysiology. The corresponding stereotactic coordinates relative to the anterior commissure in each animal (AP: anterior posterior; ML: mediolateral). Recorded thalamic neurons were from a region of the thalamus with monosynaptic connections to DMFC (inset: antidromically activated spikes in the thalamus.) Bottom: Histogram of the correlation coefficients between $e_{slow}$ inferred from the RSGP model and $z^n_{Th}$ (projection of thalamic population activity on drift direction) across recording sessions. Note that some correlations are negative because of cross-validation. Black bars correspond to the sessions in which the correlation was significantly positive (**$p<0.01$; hypothesis test by shuffling trial orders). The average correlation across all sessions and the average of those with significantly positive correlations are shown by gray and black triangles, respectively. (C) Same as B for DMFC. (D) Same as B for the caudate.

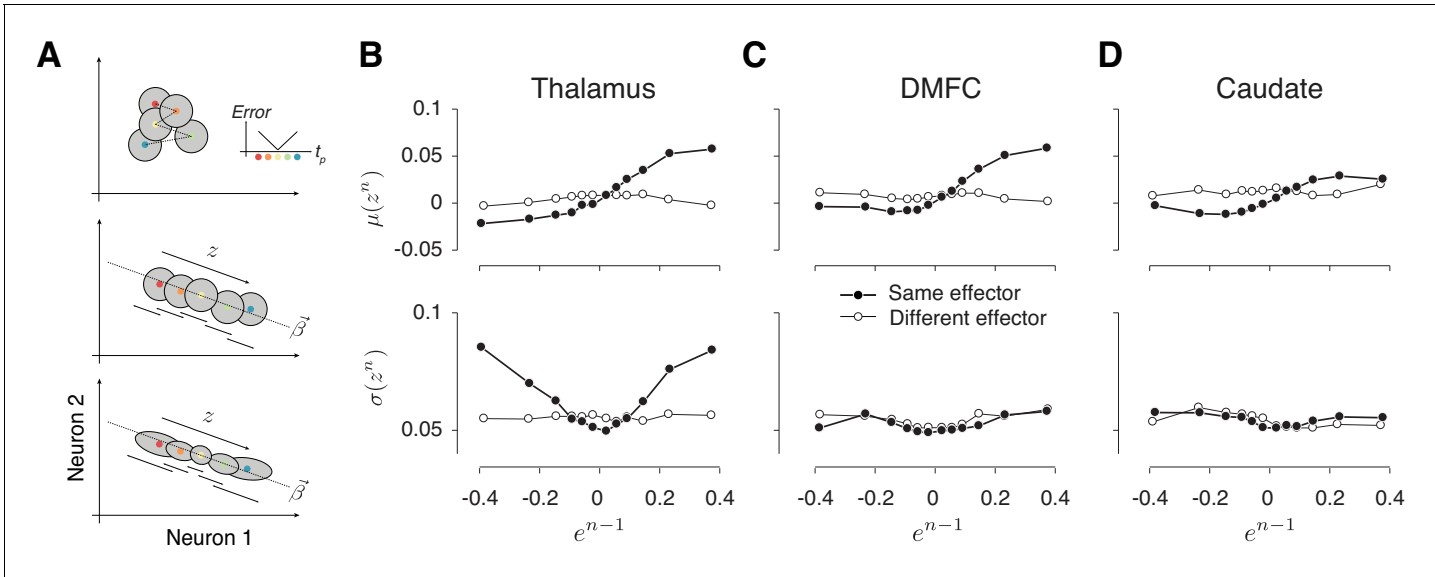

**Figure 8.** Alignment of reward-dependent neural variability and drift in thalamus, but not in DMFC and caudate. (A) Various hypotheses for how population neural activity could be related to produced intervals ($t_p$) shown schematically in two dimensions (two neurons). Top: The average neural activity (colored circles) is not systematically organized with respect to $t_p$, and the trial-by-trial variability of spike counts for a given $t_p$ around the mean (gray area) is not modulated by the size of the error. The inset shows how error increases as $t_p$ moves away from the target interval ($t_t$). Middle: Projection of average activity along a specific dimension (dotted line) is systematically ordered with respect to $t_p$, but the variability (small stacked arrows) does not depend on the size of the error. Bottom: Projection of average activity along a specific dimension is systematically ordered with respect to $t_p$ and the variability along the same axis increases with the size of error. (B) The relationship between neural activity in the thalamus on trial $n$ and relative error in the preceding trial ($e^{n-1}$). Top: Expected mean of population activity on trial $n$ ($\mu(z^n)$) along the drift direction ($\boldsymbol{\beta}$) as a function of $e^{n-1}$. Bottom: Expected standard deviation of population activity along the drift direction on trial $n$ ($\sigma(z^n)$) as a function of $e^{n-1}$. Results are shown in the same format as in *Figure 3B* (thick lines: same effector; thin lines: different effectors). (C) and (D) Same as (B) for population activity in DMFC and caudate. See *Figure 8—figure supplement 1* for the result of individual animals. The results in panels B-D were based on spike counts in a fixed 250-ms time window before Set. *Figure 8—figure supplement 4* shows that these characteristics were evident throughout the trial. *Figure 8* and *Figure 8—figure supplement 5* show the effects for each effector and that the effect for different effectors were associated with different patterns of activity across the same population of neurons.

The online version of this article includes the following figure supplement(s) for figure 8:

**Figure supplement 1.** The relationship between neural activity on trial $n$ to relative error in the preceding trial ($e^{n-1}$) in three brain areas and two animals.

**Figure supplement 2.** Drift and reward-dependent variability in the three brain areas inferred from comparable numbers of simultaneously recorded neurons.

**Figure supplement 3.** Drift and variability of spike count along the direction that decodes the produced interval ($t_p$) in the thalamus, DMFC, and caudate.

**Figure supplement 4.** Reward-dependent neural variability throughout the trial.

**Figure supplement 5.** Analysis of neural data in relation to saccade direction.

**Figure supplement 6.** Analysis of speed variability across single trials as a function of reward.

**Table 3.** Regression model fits relating spike count along the drift direction on trial $n$ ($z^n$) to error in trial $n-1$ ($e^{n-1}$).

$m_0$ and $m_1$ are parameters of the linear regression model relating the mean of $z^n$ ($\mu(z^n)$) to $e^{n-1}$, that is, $\mu(z^n) = m_0 + m_1 e^{n-1}$. $s_0$, $s_1$ and $s_2$ are parameters of the quadratic regression model relating the standard deviation of $z^n$ ($\sigma(z^n)$) to $e^{n-1}$, that is, $\sigma(z^n) = s_0 + s_1 e^{n-1} + s_2 (e^{n-1})^2$. Fit parameters are shown separately for the thalamus, DMFC and caudate and further separated depending on whether trial $n-1$ and $n$ were of the same or different effectors. Bold values for $m_1$ and $s_2$ were significantly positive (** $p<0.01$, 1% and 99% confidence intervals of the estimation were shown).

| Brain area | Parameters | | Same effector | Different effector |
|---|---|---|---|---|
| Thalamus | $\mu(z^n)$ | $m_1$ | **0.13** [0.090 0.18] | 0.0038 [-0.025 0.032] |
| | | $m_0$ | 0.008 [0.001 0.016] | 0.0041 [-0.001 0.009] |
| | $\sigma(z^n)$ | $s_2$ | **0.26** [0.19 0.34] | 0.0031 [-0.031 0.037] |
| | | $s_1$ | 0.005 [-0.013 0.023] | −0.001 [-0.01 0.0067] |
| | | $s_0$ | 0.06 [0.052 0.059] | 0.058 [0.057 0.0597] |
| DMFC | $\mu(z^n)$ | $m_1$ | **0.11** [0.0334 0.18] | −0.0027 [-0.022 0.017] |
| | | $m_0$ | 0.012 [-0.0003 0.024] | 0.0069 [0.0036 0.01] |
| | $\sigma(z^n)$ | $s_2$ | 0.071 [-0.003 0.12] | 0.0051 [-0.037 0.047] |
| | | $s_1$ | 0.0085 [-0.0022 0.019] | −0.0004 [-0.01 0.009] |
| | | $s_0$ | 0.046 [0.044 0.048] | 0.0487 [0.047 0.051] |
| Caudate | $\mu(z^n)$ | $m_1$ | **0.068** [0.016 0.12] | 0.007 [-0.016 0.030] |
| | | $m_0$ | 0.0068 [-0.002 0.015] | 0.018 [0.014 0.022] |
| | $\sigma(z^n)$ | $s_2$ | 0.069 [-0.005 0.14] | −0.002 [-0.064 0.060] |
| | | $s_1$ | 0.0006 [-0.009 0.011] | 0.001 [-0.013 0.015] |
| | | $s_0$ | 0.048 [0.046 0.050] | 0.048 [0.045 0.051] |

direction in the population activity (*Figure 8A*, bottom); otherwise, this strategy will not be able to effectively counter the degrading effect of memory drift in a context-dependent manner. In thalamus, $\sigma(z^n_{Th})$ exhibited the characteristic U-shape profile with respect to $e^{n-1}$ (*Figure 8B* bottom), which we verified quantitatively by comparing the variance of $z^n_{Th}$ for rewarded and unrewarded trials ($p<0.01$, two-sample F-test for equal variances on $z^n_{Th}$, $F(5608,4266) = 1.08$ for negative $e^{n-1}$ and $p<0.001$ for positive $e^{n-1}$, $F(4723,6125) = 1.28$), and by testing the profile using quadratic regression (see Materials and methods; *Table 3*).

As an important control, we performed the same analysis between consecutive trials associated with different effectors and we found no significant relationship between $\sigma(z^n_{Th})$ and $e^{n-1}$ ($p=0.91$, two-sample F-test for equal variances, $F(5332,3984) = 1.0$ for the negative $e^{n-1}$; $p=0.97$ for the positive $e^{n-1}$, $F(4234,5857) = 0.99$). Results did not change when we repeated these analyses after square-root transformation of spike counts to account for Poisson-like variability. Taken together, these results indicate that variability of thalamic responses was modulated by reward, and that this modulation was aligned to the drift direction.

Unlike the thalamus, in DMFC and caudate, variability along the memory drift was relatively independent of $e^{n-1}$ (*Figure 8C* bottom, 8D bottom), that is, $\sigma(z^n_{DMFC})$ and $\sigma(z^n_{Cd})$ were not significantly different after rewarded and unrewarded trials (two-sample F-test for equal variances, $F(6244,4818) = 0.87$, $p=0.99$ for the negative $e^{n-1}$; $F(4825,7572) = 1.002$, $p=0.021$ for the positive $e^{n-1}$, see *Figure 8—figure supplement 1* for each animal separately). We verified the absence of a U-shape profile using quadratic regression (see Materials and methods; *Table 3*) and ensured that the lack of modulation in DMFC and caudate compared to the thalamus was not because of a difference in the number of simultaneously recorded neurons (*Figure 8—figure supplement 2*). Together, these results provide evidence that the effect of reward in DMFC and caudate was not aligned to the $\beta$ associated with the slow fluctuations. We note, however, that an unconstrained decoding strategy aimed at simultaneously capturing both components of errors could find directions along which the slow fluctuations and the effect of reward are aligned in all three areas (*Figure 8—figure supplement 3*, *Table 4*).

**Table 4.** Regression model fits relating spike count along the direction that predicts produced interval ($t_p$) on trial $n$ ($z^n$) to error in trial $n-1$ ($e^{n-1}$).

$m_0$ and $m_1$ are parameters of the linear regression model relating the mean of $z^n$ ($\mu(z^n)$) to $e^{n-1}$; i.e., $\mu(z^n) = m_0 + m_1 e^{n-1}$. $s_0$, $s_1$ and $s_2$ are parameters of the quadratic regression model relating the standard deviation of $z^n$ ($\sigma(z^n)$) to $e^{n-1}$; i.e., $\sigma(z^n) = s_0 + s_1 e^{n-1} + s_2 (e^{n-1})^2$. Fit parameters are shown separately for the thalamus, DMFC and caudate and further separated depending on whether trial $n-1$ and $n$ were of the same or different effectors. Bold indicated significantly positive value (p<0.01, 1% and 99% confidence intervals of the estimation were shown).

| Brain area | Parameters | | Same effector | Different effector |
|---|---|---|---|---|
| Thalamus | $\mu(z^n)$ | $m_1$ | **0.13** [0.096 0.17] | **0.018** [0.002 0.036] |
| | | $m_0$ | 0.012 [0.005 0.019] | 0.006 [0.002 0.01] |
| | $\sigma(z^n)$ | $s_2$ | **0.32** [0.19 0.44] | −0.019 [-0.07 0.066] |
| | | $s_1$ | 0.012 [-0.019 0.046] | −0.003 [-0.02 0.014] |
| | | $s_0$ | 0.063 [0.055 0.071] | 0.065 [0.061 0.069] |
| DMFC | $\mu(z^n)$ | $m_1$ | **0.091** [0.043 0.14] | 0.011 [-0.0073 0.29] |
| | | $m_0$ | 0.014 [0.0046 0.024] | 0.011 [0.0075 0.015] |
| | $\sigma(z^n)$ | $s_2$ | **0.15** [0.087 0.21] | −0.017 [-0.071 0.036] |
| | | $s_1$ | −0.006 [-0.022 0.009] | 0.001 [-0.0041 0.024] |
| | | $s_0$ | 0.049 [0.045 0.052] | 0.051 [0.047 0.054] |
| Caudate | $\mu(z^n)$ | $m_1$ | **0.077** [0.036 0.12] | 0.007 [-0.012 0.027] |
| | | $m_0$ | 0.010 [0.0014 0.018] | 0.015 [0.011 0.019] |
| | $\sigma(z^n)$ | $s_2$ | **0.16** [0.11 0.21] | **0.044** [0.016 0.072] |
| | | $s_1$ | 0.005 [-0.007 0.17] | −0.0025 [-0.01 0.0047] |
| | | $s_0$ | 0.050 [0.047 0.053] | 0.051 [0.049 0.053] |

# Discussion

Variability is a ubiquitous property of behavior that can either degrade performance or promote learning through exploration. Here, we were able to advance our understanding of the function and neurobiology of variability in three directions. First, we found an important role for memory drifts in timing variability and showed that humans and monkeys use reward to calibrate memory against such drifts. Second, using model-based analysis of behavior, we showed how a combination of directed and random explorations promoted by trial outcome can manifest as modulations of behavioral variability. Finally, we characterized the neural underpinnings of variability associated with memory drifts and trial outcome across populations of neurons within the cortical-basal ganglia circuits.

## Role of memory drift in behavioral variability

A key feature of the behavioral data was the presence of slow fluctuations in produced intervals leading to serial correlations extending over minutes and across dozens of trials. These fluctuations have been reported in various behavioral tasks (*Gilden et al., 1995*; *Chen et al., 1997*; *Murakami et al., 2017*) and are typically attributed to fatigue, arousal, or other nonspecific factors modulating internal states (*Luck et al., 1997*; *Niell and Stryker, 2010*; *Harris and Thiele, 2011*; *Kato et al., 2012*; *Lee and Dan, 2012*; *Vinck et al., 2015*). Although global internal state modulations are likely present in our experiment, they cannot be the only driver since the serial correlations were strongly context-specific. Based on these observations, we reasoned that these fluctuations may reflect drifts in memory. This interpretation was consistent with the results of our control experiment showing diminished serial correlations when memory demands were reduced (*Figure 2—figure supplement 1D*). However, further work is needed to fully characterize the nature of these slow fluctuations. For example, the slow component may be in part a reflection of an active averaging process to maintain a stable memory (*Joiner and Smith, 2008*) as suggested by error-based motor learning studies (*Smith et al., 2006*; *Wolpert et al., 2011*; *Huberdeau et al., 2015*). Indeed, behavioral results in the probabilistic reward experiment (*Figure 5—figure supplement 2A*) as well as corresponding fits the RSGP model (*Figure 5—figure supplement 2B*) suggest that slow fluctuations may in part be controlled by reward-dependent exploratory behavior.

A more puzzling observation was the specificity of memory drifts with respect to the target interval for the same effector. We currently do not have a definite explanation for this result, but our previous work in the domain of time interval production (*Wang et al., 2018*) and reproduction (*Remington et al., 2018b*) as well as other studies in the motor system (*Afshar et al., 2011*; *Ames et al., 2014*; *Sheahan et al., 2016*; *Hauser et al., 2018*; *Vyas et al., 2018*) suggest that several aspects of movement control can be understood in terms of adjusting inputs and initial conditions of a dynamical system (*Churchland et al., 2012*; *Remington et al., 2018a*). Accordingly, the interval specificity of the memory drifts suggests that distinct patterns of neural activity set the interval-dependent input and/or initial condition, which is consistent with our previous work (*Wang et al., 2018*).

## Role of reinforcement in behavioral variability

In our task, reinforcement is the only information provided experimentally that subjects can use to calibrate their memory of the target interval. Therefore, the computational demands in our task fall squarely within the framework of RL. Most existing RL models have focused on experimental settings in which the agent faces a discrete set of options and/or a limited action space (*Daw et al., 2006*; *Hayden et al., 2011*; *Lee et al., 2011*; *Wilson et al., 2014*). In these situations, RL models posit that the agent keeps track of the value of available options and adopts a suitable policy to choose among them (*Sutton and Barto, 1998*). In our task, subjects have to use reinforcement to choose the correct interval, which is a continuous variable. In theory, RL can be extended to continuous variables, but doing so would require the brain to represent an infinite-dimensional value function, which is implausible. Moreover, in the case of motor timing, produced intervals typically differ from what was intended, and that would interfere with the agent's ability to correctly evaluate the intended action.

Due to these complications, recent studies have proposed an alternative explore-exploit strategy for continuous variables in which the agent uses reinforcement to directly regulate behavioral variability (*Tumer and Brainard, 2007*; *Fee and Goldberg, 2011*; *Wu et al., 2014*; *Pekny et al., 2015*; *Santos et al., 2015*; *Dhawale et al., 2017*). Our results were consistent with this view: variability was modulated by reward in a manner that was consistent with an explore-exploit strategy. This hypothesis was strongly supported by the results of our psychophysical experiment using a probabilistic reward schedule, which confirmed that the trial outcome had a causal effect on behavioral variability (*Figure 4*). Another key observation was that the effect of reward on behavioral variability was context-specific (*Figure 3*), that is, variability associated with producing a specific interval with a specific effector was most strongly dependent on reward history in trials of the same interval and effector. Given that the memory drifts were also context-specific, this finding indicates that one important function of reward-dependent regulation of variability is to counter memory drifts.

## Extending RL to continuous variables

When the action space is discrete as in multi-arm bandit tasks, one can straightforwardly distinguish between exploitation and exploration: exploitation is selecting options that were previously rewarded, and exploration is selecting alternatives about which the agent is uncertain. This distinction, however, is neither straightforward nor helpful when dealing with continuous variables because the state space is too large and all options engender uncertainty. In this scenario, exact exploitation is not possible. Instead, exploitation can be viewed as a form of directed exploration whose objective is to bias responses toward previously rewarded states. This can be contrasted with random explorations that drive responses toward less frequently visited states, and can thus be viewed as increasing variance (*Dhawale et al., 2017*).

In our experiment, the space of possible responses is continuous and individual responses are inherently variable. Therefore, it is impossible to produce the same exact time interval after a rewarded trial. Accordingly, we formulated the RL problem in terms of moderating directed versus random explorations (*Wilson et al., 2014*). To capture the effect of reward on behavior, we developed a reward-sensitive Gaussian process (RSGP) model that implements a Bayesian updating strategy that simultaneously adjusts the bias and variance. When trial outcomes are better than expected, the RSGP tips the balance toward directed exploration by adjusting the bias, and when the reward drops, it increases variance.

RSGP was able to capture both the long-term effect of memory drift, and the short-term effect of reward on variability. It also captured behavioral observations in the probabilistic reward experiment where we validated the causal effect of reward on variability (*Figure 5—figure supplement 2B*). A recent motor learning study in rodents found that the effect of reward on behavioral variability may last a few trials into the future (*Dhawale et al., 2019*). This was also evident in our data (*Figure 2—figure supplement 1D–E*), and RSGP provided a quantitative measure of the temporal extent of this effect (*Figure 5B*).

More generally, the RSGP suggests that categorical distinctions such as exploration versus exploitation or directed versus random explorations might be somewhat arbitrary and ill-defined. Instead, it may be more fruitful to adopt an alternative viewpoint in which the effect of reinforcement is quantified in terms of how trial outcome alters the distribution of future responses (e.g. both the mean and variance). As such, we expect RSGP to help future studies quantify the strength and persistence with which reinforcement guards against ubiquitous non-stationarities in behavior (*Weiss et al., 1955*; *Merrill and Bennett, 1956*; *Laming, 1979*; *Gilden et al., 1995*; *Chen et al., 1997*; *Chaisanguanthum et al., 2014*; *Murakami et al., 2017*).

## Memory drift in the cortico-basal ganglia circuits

Behavioral variability in timing tasks is thought to have a multitude of distributed biophysical and synaptic origins (*Gibbon et al., 1984*; *Mauk and Buonomano, 2004*; *Paton and Buonomano, 2018*). Several studies have been able to trace this variability in vivo to spiking activity of neurons in cortico-basal ganglia circuits (*Murakami et al., 2014*; *Murakami et al., 2017*; *Gouvêa et al., 2015*; *Dhawale et al., 2017*; *Merchant and Averbeck, 2017*; *Wang et al., 2018*). We previously found that a circuit comprised of DMFC, DMFC-projecting thalamus and caudate plays a causal role in the control of movement initiation time (*Wang et al., 2018*) and that certain patterns of activity in each area were correlated with behavioral variability on a trial-by-trial basis. Here, we additionally confirmed that population activity in these areas carries a signal correlated with slow fluctuations of behavior. This finding is broadly consistent with previous studies reporting correlates of internal state changes and/or slow behavioral fluctuations in the thalamus (*Halassa et al., 2014*), the medial frontal cortex (*Narayanan and Laubach, 2008*; *Sul et al., 2010*; *Karlsson et al., 2012*; *Murakami et al., 2017*), and the caudate (*Lauwereyns et al., 2002*; *Lau and Glimcher, 2007*; *Santos et al., 2015*). However, the effector- and interval-dependent nature of these fluctuations in our data suggests that they may partially reflect context-specific memory drifts. The key feature of our task that enabled us to uncover this specificity was the alternation between different contexts on a trial-by-trial basis. Since many previous studies did not include this feature, it is possible that certain aspects of neural variability previously attributed to nonspecific internal state changes were in part caused by memory drifts related to specific task rules and parameters. Indeed, drifts and degradation of instrumental memories may be a key limiting factor in motor skill performance (*Ajemian et al., 2013*).

Although we found a neural correlate of these drifts in all three brain areas, we cannot make a definitive statement about the loci of the underlying synaptic and biophysical drifts. It is likely that the memory has a distributed representation, in which case the drift may result from stochastic processes distributed across multiple brain areas. It is also possible that different contexts engage specific sub-circuits such as corticostriatal synapses (*Fee and Goldberg, 2011*; *Xiong et al., 2015*), and circuit-level interactions cause these drift to be present in other nodes of the cortico-basal ganglia circuit.

For different effectors, context-specificity may be attributed to different patterns of activity across the same population of neurons or by distinct populations of neurons. Our results in the cortico-basal ganglia circuits provided evidence for the former (*Figure 8*, *Figure 8—figure supplement 5*). However, we cannot rule out the possibility that this context-specificity arises partially from execution noise within distinct effector-specific downstream areas (e.g. brainstem) as we did not record from those areas. However, execution noise is generally considered to be irreducible (*Faisal et al., 2008*; *Dhawale et al., 2017*) and is therefore not expected to be influenced by reinforcement. Previous work suggests that central noise (e.g. in the cortico-basal ganglia circuits) plays a significant role in output variability (*Churchland et al., 2006*; *van Beers, 2009*). It is this portion of variability that is likely subject to adjustments through reinforcement.

## Reinforcement via thalamus

Next, we asked whether the variability of neural activity in the thalamus, DMFC, and caudate was modulated by reward in the preceding trial in the same context-dependent manner as in the behavior. According to our hypothesis, the key function of the reward-dependent regulation of variability is to counter the memory drifts. This hypothesis makes a specific prediction: reward should modulate the specific pattern of population neural activity that corresponds to memory drifts in the behavior, which we referred to as drift direction. Analysis of neural activity revealed that this effect was present in the thalamus but not in DMFC or caudate. In the thalamus, spike count variability along the drift direction increased after rewarded trials and decreased after unrewarded trials in a context-specific manner. Previous studies have reported that in the thalamus firing rates are modulated on a trial-by-trial basis by attention (*McAlonan et al., 2008*; *Saalmann et al., 2012*; *Zhou et al., 2016*) and rule/context-dependent computations (*Schmitt et al., 2017*; *Wang et al., 2018*). Our work demonstrates that modulation of thalamic activity may additionally subserve reward-based calibration of movement initiation times. It will be important for future studies to investigate whether this finding generalizes to other movement parameters.

The fact that the same effect was not present in DMFC and caudate serves as a negative control and thus strengthens our conclusions. However, this begs the question of why this regulation was not inherited by DMFC and caudate, especially given that DMFC receives direct input from the region of the thalamus we recorded from. The answer to this question depends on the nature of signal transformations along the thalamocortical pathway. While some experiments have suggested similar response properties for thalamus and their cortical recipients (*Sommer and Wurtz, 2006*; *Guo et al., 2017*), others have found that thalamic signals may undergo specific transformations along the thalamocortical pathway (*Hubel and Wiesel, 1962*; *Berman and Wurtz, 2011*; *Wimmer et al., 2015*; *Schmitt et al., 2017*; *Wang et al., 2018*). Therefore, the extent to which we should expect responses in DMFC and thalamus to have similar properties is unclear.

Our approach for investigating whether the effect of reinforcement was aligned with memory drift was correlative: we used a cross-validated decoding strategy to infer the response pattern most strongly associated with memory drift and tested whether reinforcement exerted its effect along that pattern. According to this analysis, only in the thalamus the two effects were appropriately aligned. However, we cannot rule out the possibility that behavioral control may be mediated by other patterns of activity in the DMFC. Indeed, with a complementary analysis using an unconstrained decoding approach, we were able to find patterns of activity in all three brain areas that simultaneously reflected the effects of memory drift (*Figure 7*) and reinforcement (*Figure 8—figure supplement 3*). Therefore, an important consideration for future work is to use perturbation methods to causally verify the direction in the state space associated with memory drift and reinforcement.

However, other considerations are consistent with thalamus playing a strong role in controlling timing variability. For example, it has been shown that the inactivation of thalamus has a much stronger effect on modulating animals' motor timing variability compared to DMFC and caudate (*Wang et al., 2018*). Moreover, the nature of signals in DMFC-projecting thalamus and DMFC during motor timing are different: DMFC neurons have highly heterogeneous response profiles that evolve at different speeds depending on the interval, whereas thalamic neurons carr signals whose strength (i.e. average firing rate) encode the underlying speed. This transformation may provide an explanation for why reward-dependent modulation of firing rates was evident in the thalamus but not in DMFC. As thalamic neurons encode the interval in their average firing rates, it is expected that regulation of timing variability by reward would similarly impact average firing rates. In contrast, in DMFC, the key signal predicting behavior was the speed at which neural trajectories evolved over time – not the neural states along the trajectory. This predicts that reward should alter the variability of the speed of neural trajectories. In principle, it is possible to verify this prediction by estimating the variance of the speed of neural trajectories as a function of reward. However, this estimation is challenging for two reasons. First, speed in a single trial is derived from changes in instantaneous neural states, and the estimation of instantaneous neural states is unreliable unless the number of recorded neurons exceeds the dimensionality of the subspace containing the neural trajectory (*Gao, 2017*). Second, our predictions are about the variance – not mean – of speed, and estimating

variance adds another layer of statistical unreliability unless the number of neurons or trials are sufficiently large.

Nonetheless, we devised a simple analysis to estimate the variance of the speed of neural trajectories across single trials in all three areas (*Figure 8—figure supplement 6*). As predicted by our hypothesis, the effect of reward on neural activity in the thalamus was different from that in the DMFC and caudate. In thalamus, the reward adjusted the variance of average firing rates, but not the variance of speed . In contrast, in the DMFC and caudate, the reward modulated the variance of the speed at which neural trajectories evolve. Moreover, these effects were present only for consecutive trials associated with the same effector. These results further substantiate our hypothesis that reward regulates variability by adjusting the average firing rates in thalamus, and that this effect leads to the control of the variance of the speed at which neural trajectories evolve in the DMFC and caudate.

One open question pertains to which brain areas supply the relevant information for the reward-dependent control of behavioral variability. One salient example where cortical variability is adjusted rapidly and in a behaviorally relevant fashion is in the domain of attentional control where spatial cueing can lead to a drop of correlated variability in spiking across sensory cortical neurons whose receptive fields correspond to the cued location (*Cohen and Maunsell, 2009*; *Mitchell et al., 2009*; *Ruff and Cohen, 2014*; *Ni et al., 2018*). While we do not know which areas might directly control motor timing variability, we note that the area of thalamus we have recorded from receives information from three major sources, the frontal cortex, the output nuclei of the basal ganglia, and the deep nuclei of the cerebellum (*Middleton and Strick, 2000*; *Kunimatsu et al., 2018*). Modulation of variability prior to movement initiation has been reported in motor and premotor areas (*Churchland et al., 2006*; *Churchland et al., 2010*), and distinct correlates of explore-exploit strategy have been found across multiple nodes of the frontal cortex (*Hayden et al., 2011*; *Tervo et al., 2014*; *Ebitz et al., 2018*; *Massi et al., 2018*; *Sarafyazd and Jazayeri, 2019*). Therefore, it is possible that modulation of variability in thalamus originates from correlated variability in the frontal cortex. To act as an effective learning mechanism, such correlated variability must be additionally sensitive to reward-dependent neuromodulatory signals such as dopamine (*Frank et al., 2009*) possibly by acting on local inhibitory neurons (*Huang et al., 2019*). The basal ganglia could also play a role in reward-dependent control of thalamic firing rates (*Kunimatsu and Tanaka, 2016*; *Kunimatsu et al., 2018*). For example, single neuron responses in substantia nigra pars reticulata that were strongly modulated by reward schedule (*Yasuda and Hikosaka, 2015*) can influence neural responses in the thalamus. Finally, the cerebellum plays a central role in trial-by-trial calibration of motor variables (*Ito, 2002*; *Medina and Lisberger, 2008*; *Herzfeld et al., 2015*) including movement initiation time (*Ashmore and Sommer, 2013*; *Kunimatsu et al., 2018*; *Narain et al., 2018*) and thus is a natural candidate for calibrating firing rates in thalamus, although how such calibration could be made reward-sensitive remains an open question (*Hoshi et al., 2005*). In sum, our work provides behavioral, modeling, and neurophysiological evidence in support of the hypothesis that the brain uses reinforcement to regulate behavioral variability in a context-dependent manner.

## Materials and methods

Two adult monkeys (*Macaca mulatta*; one female, one male) and five human subjects (18–65 years, two females and three males) participated in the main task. Two additional monkeys (both male) participated in the memory control experiment. In addition, five more human subjects (18–65 years, two females and three males) participated in the probabilistic reward task to test the causal effect of reward on variability. The Committee of Animal Care at Massachusetts Institute of Technology approved all animal experiments. The Committee on the Use of Humans as Experimental Subjects at Massachusetts Institute of Technology approved all human experiments. As per our approved protocol, all human participants provided consent for the use and publication of data prior to data collection. All procedures conformed to the guidelines of the National Institutes of Health.

### Animal experiments

Monkeys were seated comfortably in a dark and quiet room. The MWorks software package (https://mworks.github.io) running on a Mac Pro was used to deliver stimuli and to control behavioral contingencies. Visual stimuli were presented on a 23-inch monitor (Acer H236HL, LCD) at

a resolution of 1920 × 1080, and a refresh rate of 60 Hz. Auditory stimuli were played from the computer's internal speaker. Eye position was tracked with an infrared camera (Eyelink 1000; SR Research Ltd, Ontario, Canada) and sampled at 1 kHz. A custom-made manual button, equipped with a trigger and a force sensor, was used to register button presses.

## The Cue-Set-Go task

Behavioral sessions in the main experiment consisted of four randomly interleaved trial types in which animals had to produce a target interval ($t_t$) of either 800 ms (Short) or 1500 ms (Long) using either a button press (Hand) or a saccade (Eye). The trial structure is described in the main Results (*Figure 1A*). Here, we only describe the additional details that were not described in the Results. The 'Cue' presented at the beginning of each trial consisted of a circle and square. The circle had a radius of 0.2 deg and was presented at the center of the screen. The square had a side of 0.2 deg and was presented 0.5 deg below the circle. For the trial to proceed, the animal had to foveate the circle (i.e. eye fixation) and hold its hand gently on the button (i.e. hand fixation). The animal had to use the hand contralateral to the recorded hemifield. We used an electronic window of 2.5 deg around the circle to evaluate eye fixation, and infrared emitter and detector to evaluate hand fixation. After 500–1500 ms delay period (uniform hazard), a saccade target was flashed eight deg to the left or right of the circle. The saccade target ('Tar') had a radius of 0.25 deg and was presented for 250 ms. After another 500–1500 ms delay (uniform hazard), an annulus ('Set') was flashed around the circle. The Set annulus had an inner and outer radius of 0.7 and 0.75 deg and was flashed for 48 ms. Trials were aborted if the eye moved outside the fixation window or hand fixation was broken before Set.

For responses made after Set, the produced interval ($t_p$) was measured from the endpoint of Set to the moment the saccade was initiated (eye trial) or the button was triggered (hand trial). Reward was provided if the animal used the correct effector and $t_p$ was within an experimentally controlled acceptance window. For saccade responses, reward was not provided if the saccade endpoint was more than 2.5 deg away from the extinguished saccade target, or if the saccade endpoint was not acquired within 33 ms of exiting the fixation window.

## Reward function for monkeys in the main task

The reward was determined by a truncated triangular function of error ($t_p$-$t_t$). The maximum reward was 0.5 ml of juice when $t_p = t_t$. The reward dropped linearly for larger errors so long error was smaller than an acceptance window. For errors larger than the acceptance window, no reward was provided. The acceptance window was adjusted adaptively on a trial-by-trial basis. After each rewarded trial, the acceptance window was made smaller, and after each unrewarded trial, the window was made larger. This so-called one-up-one-down staircase procedure ensured that approximately 50% of the trials were rewarded. The acceptance window for the four trial conditions was set by independent staircases. The change in the size of the acceptance window after each trial was set to 8 and 15 ms for the 800 and 1500 ms target intervals, respectively. As a result of this procedure, animals received reward on nearly half of trials (57% in monkey A and 51% in monkey D). Visual and auditory feedback accompanied rewarded trials. For the visual feedback, either the color of the saccade target (for Eye trials) or the central square (for the Hand trials) turned green. The auditory feedback was one of the computer's default tones.

## No-memory control task

To validate our hypothesis that slow fluctuations in animals' behavior arose from memory fluctuations, we performed a control experiment in two naive monkeys. In the control experiment, the animals did not have to remember the target interval $t_t$, but instead measured it on every trial. This was done by presenting an additional flash ('Ready') shortly before the Set flash such that the interval between Ready and Set was fixed and equal to $t_t$. This effectively removed the need for the animal to hold the target interval in memory. We limited the control experiment to a single effector (Eye) and a single interval ($t_t$ = 840 ms). The reward was also determined by a truncated triangular function similar to the main task.

## Electrophysiology

Recording sessions began with an approximately 10 min warm-up period to allow animals to recalibrate their timing and exhibit stable behavior. We recorded from 932 single- or multi-units in the thalamus, 568 units in the dorsomedial frontal cortex (DMFC), and 509 units in caudate, using 24-channel linear probes with 100 μm or 200 μm interelectrode spacing (V-probe, Plexon Inc). The number of simultaneously recorded neurons across sessions were shown in *Figure 8—figure supplement 2A* (mean ±s.d.; Thalamus: 17.9 ± 9.0; DMFC: 9.2 ± 4.3; Caudate: 14.8 ± 6.5). The DFMC comprises supplementary eye field, dorsal supplementary motor area (i.e. excluding the medial bank), and pre-supplementary motor area. We recorded from the left hemisphere from Monkey A and right from Monkey D, which was contralateral to the preferred hand used in the button press. Recording locations were selected according to stereotaxic coordinates with reference to previous studies as well as each animal's structural MRI scan. The region of interest targeted in the thalamus was within 1 mm of antidromically identified neurons in the medial portion of the lateral thalamus, also known as Area X. All behavioral and electrophysiological data were timestamped at 30 kHz and streamed to a data acquisition system (OpenEphys). Spiking data were bandpass filtered between 300 Hz and 7 kHz, and spike waveforms were detected at a threshold that was typically set to three times the RMS noise. Single- and multi-units were sorted offline using custom software, MKsort (*Kaufman, 2013*).

## Antidromic stimulation

We used antidromic stimulation to localize DMFC-projecting thalamic neurons. Antidromic spikes were recorded in response to a single biphasic pulse of duration 0.2 ms (current <500 uA) delivered to DMFC via low impedance tungsten microelectrodes (100–500 KΩ, Microprobes). A stainless-steel cannula guiding the tungsten electrode was used as the return path for the stimulation current. Antidromic activation evoked spikes reliably at a latency ranging from 1.8 to 3 ms, with less than 0.2 ms jitter.

## Human experiments

### The Cue-Set-Go task

Each experimental session lasted approximately 60 min. Each subject completed 2–3 sessions per week. Similar to monkeys, experiments were conducted using the MWorks. All stimuli were presented on a black background monitor. Subjects were instructed to hold their gaze on a fixation point and hold a custom-made push button using their dexterous hand, throughout the trial. Subjects viewed the stimuli binocularly from a distance of approximately 67 cm on a 23-inch monitor (Apple, A1082 LCD) driven by a Mac Pro at a refresh rate of 60 Hz in a dark and quiet room. Eye positions were tracked with an infrared camera (Eyelink 1000 plus, SR Research Ltd.) and sampled at 1 kHz. The state of the button was converted and registered as digital TTL through a data acquisition card (National Instruments, USB-6212). The Cue-Set-Go task for humans was similar to monkeys with the following exceptions: (1) in each session, we used a single $t_t$ sampled from a normal distribution (mean: 800 ms, std: 80 ms); (2) the saccadic target was 10 deg (instead of 8 deg) away from the fixation point; (3) humans only received visual and auditory feedback (an no physical reward).

### Reward function for humans in the main task

The feedback contingencies were identical to monkeys with the only difference that humans only received visual and auditory feedback and no reward. The one-up one-down staircase procedures led to an average of 50.2% trials with positive feedback. For the positive feedback, the color of the saccade target (for Eye trials) or the central square (for Hand trials) turned green. The auditory cue used two different tones for positive and negative feedback.

### The probabilistic reward task

A set of different human subjects performed the same timing interval production task, except that the nature of the feedback was provided probabilistically. Subjects received 'correct' feedback with the probability of 0.7 when $t_p$ was within a window around $t_t$, and with the probability of 0.3 when errors were outside that window (*Figure 4C*). The window length was adjusted based on the overall behavioral variability so that each subject receives approximately 50% 'correct' feedback in each

behavioral session. Only one effector - hand pressing - was used in this causal experiment, as we have established that both the slow and reward regulated variability were effector specific. All the experimental parameters were set identical to the main experiment.

## Data analysis

All offline data processing and analyses were performed in MATLAB (2019b, MathWorks Inc).

### Analysis of behavior

Behavioral data for the CSG task comprised of N = 203 behavioral sessions consisting of n = 167,115 trials in monkeys (N = 95, n = 71,053 for monkey A and N = 108, n = 96,062 for monkey D), N = 62 sessions and n = 59,297 trials in humans, and N = 51 sessions and n = 30,695 trials in the probabilistic feedback experiment in human subjects. Behavioral data for the no-memory control task was collected in N = 26 sessions consisting of n = 75,652 trials in two naive monkeys (N = 9, n = 32,041 for monkey G and N = 17, n = 43,611 for monkey H).

We computed the mean and standard deviation of $t_p$, denoted by $\mu(t_p)$ and $\sigma(t_p)$, respectively, for each trial type within each session (*Figure 1C*). We additionally analyzed local fluctuations of $\mu(t_p)$ and $\sigma(t_p)$ by computing these statistics from running blocks of 50 trials within session and averaged across sessions. The mean of $\sigma(t_p)$ for each corresponding $\mu(t_p)$ bin and the averaged reward across all trials in each $\mu(t_p)$ bin were plotted in *Figure 1D*. Results were qualitatively unchanged when the block length was increased or decreased by a factor of 2.

We also examined the slow fluctuations of $t_p$ for pairs of trials that were either of the same type (e.g. Eye-Short versus Eye-Short) or of different types (e.g. Hand-Long versus Eye-Short). For trials of the same type, we computed partial correlation coefficients of $t_p$ pairs by fitting a successive autoregressive model with the maximum order of 60 trial lag (*Box, 2015*; *Figure 2A*).1 and 99% confidence bounds were estimated at 2.5 times the standard deviation of the null distribution. For trials of different types, we calculated the Pearson correlation coefficient of pairs of $t_p$ of various lags. To clarify our analysis, we use an example of how we estimated the cross correlation between pairs of HS-ES with a trial lag of 10: (1) normalize (z-score) two $t_p$ vectors associated with HS and ES in each session; (2) take pairs of HS-ES that are 10 trials apart within each session; (3) combine the pairs across sessions; (4) compute Pearson correlation coefficient. We also computed a corresponding null distribution from 100 randomly shuffled trial identity. 1 and 99% confidence intervals were estimated from the null distribution.

Finally, we quantified the mean and standard deviation of the relative error denoted by $\mu(e^n)$ and $\sigma(e^n)$ as a function of error in the previous trial ($e^{n-1}$) for each pair of trial types (*Figure 3—figure supplement 1*). Deriving reliable estimates of $\mu(e^n)$ and $\sigma(e^n)$ as a function of $e^{n-1}$ from a non-stationary process requires a large number of trials. Since each trial can be of four different types (ES, EL, HS, HL), consecutive trials comprise 16 distinct conditions (e.g. ES-EH, HL-EL, etc.). The limited number of trials in each session limited the reliability of statistics estimated for each individual condition. To gain more statistical power, we combined results across trial types in two ways. First, for each effector, we combined the Short and Long trial types by normalizing $t_p$ values by their respective $t_t$. The resulting variable was defined as relative error $e^n = (t_p^n - t_t)/t_t$. This reduced the number of conditions by a factor of four, leaving consecutive trials that were either associated with the same effector or with different effectors (e.g. E-E, E-H, H-E, and H-H). We further combined trials to create a 'same effector' condition that combined E-E with H-H, and a 'different effector' condition that combined E-H with H-E. Animals and human subjects were allowed to take breaks during the experimental sessions. However, the pairs of consecutive trials used in all analyses, regardless of the trial condition, were restricted to the two consequent and completed trials that were no more than 7 s apart.

We examined the relationship between $\mu(e^n)$ and $e^{n-1}$ using a linear regression model of the form $\mu(e^n) = m_0 + m_1 e^{n-1}$, and the relationship between $\sigma(e^n)$ and $e^{n-1}$ using a quadratic regression model of the form $\sigma(e^n) = s_0 + s_1 e^{n-1} + s_2 (e^{n-1})^2$. We assessed the significance of the monotonic relationship between $\mu(e^n)$ and $e^{n-1}$ by the slope of linear regression ($m_1$), and the significance of the U-shaped profile in $\sigma(e^n)$ by the square term of a quadratic regression ($s_2$). Note that the choice of these models was not theoretically motivated; we simply considered these regression models to be an approximate function for testing the profile of $\mu(e^n)$ and $\sigma(e^n)$ as a function of $e^{n-1}$. The 1 and 99%

confidence intervals were defined as $s_2 \pm SE(s_2)$, where $s_2$ is the estimated coefficient and $SE(s_2)$ is its standard error. $t_{(.01,\ N-p)}$ is the 1 percentile of t-distribution with $N-p$ degrees of freedoms.

We applied the same analysis to the probabilistic reward experiment in humans after grouping the trials based on whether the feedback in the previous trials was 'correct' or 'incorrect'. Before analyzing the variability, we removed outlier trials defined as trials in which $t_p$ was more than three standard deviations away from the mean. Since subjects were allowed to initiate the trials, we also excluded pairs of adjacent trials that were more than 7 seconds apart. When combining data across sessions, we normalized the relative error across sessions and subjects. To avoid sampling bias, the same number of trials were drawn repeatedly after 'correct' or 'incorrect'. The standard error of the mean was computed from 100 repeats with replacement (*Figure 4D*).

## Reward-sensitive Gaussian process (RSGP) model simulation and fitting

We constructed a reward-sensitive Gaussian process model whose covariance function, $K_{RSGP}$, is a weighted sum of two kernels, a traditional squared exponential kernel, for which we used subscript SE ($K_{SE}$), and a reward-sensitive kernel with subscript RS ($K_{RS}$). The two kernels contribute to $K_{RSGP}$ through scale factors $\sigma^2_{SE}$ and $\sigma^2_{RS}$, respectively. In both kernels, the covariance term between any two trials (trial $n$ and $n$-$r$) drops exponentially as a function of trial lag ($r$). The rates of drop for $K_{SE}$ and $K_{RS}$ are specified by characteristic length parameters, $l_{SE}$ and $l_{RS}$, respectively. The model also includes a static source of variance, $\sigma^2_0 I$ ($I$ stands for the identity matrix):

$$K_{RSGP}(n, n-r) = \sigma^2_{SE} K_{SE}(n, n-r) + \sigma^2_{RS} K_{RS}(n, n-r) + \sigma^2_0 I$$

$$K_{SE}(n, n-r) = \exp\left(-\frac{r^2}{2l^2_{SE}}\right)$$

$$K_{RS}(n, n-r) = \begin{cases} \exp\left(-\frac{r^2}{2l^2_{RS}}\right) & \text{if trial } n-r \text{ was rewarded} \\ 0 & \text{otherwise} \end{cases}$$

Note that $K_{RS}$ values depend on reward history and are thus not necessarily invariant with respect to time; that is, $K(n, n-i) \neq K(m, m-i)$. This formulation allows past rewarded trials to have higher leverage on future trials and this effect drops exponentially for rewarded trials farther in the past.

We simulated the RSGP by applying GP regression based on the designated covariance function (*Table 5*). To simplify our formulations and without loss of generality, we replaced $\sigma^2_{SE}$ and $\sigma^2_{RS}$ by $\alpha\sigma^2$ and $(1 - \alpha)\sigma^2$, respectively where $\alpha = 1.0, 0,$ and $0.5$ for the three examples (*Figure 5A*).

As both the slow fluctuation and reward regulation were context specific, we fit the model to behavioral data for each trial type (ES, EL, HS, and HL) separately. To do so, we ordered $t_p$ values associated with the same trial type within each behavioral session chronologically and treated them as consecutive samples from the model irrespective of the actual trial lag between them. Although this strategy made the model fitting more tractable, the inferred length constants in units of trials are likely smaller than the true values in the data. For the same reason, the temporal distance in the kernel function was different from the actual trial lag (*Figure 2B*). Methods for fitting behavioral data to the RSGP model were adapted from *Gaussian Processes for Machine Learning* (*Rasmussen and Williams, 2006*). The objective was to maximize the marginal likelihood of the observed data with respect to the hyperparameters ($l_{SE}$, $\sigma_{SE}$, $l_{RS}$, $\sigma_{RS}$, $\sigma_0$). Using simulations, we found that optimization

**Table 5.** Algorithm for generating time series based on RSGP model.

for $n = 1...N$ do
1. Given the previous value and reward history, infer the mean and
The variance of $t^n_p$ from the conditional distribution
$$\left[t^1_p, \cdots, t^{n-1}_p, t^n_p\right]^\top \sim \mathcal{N}\left(_p>, \begin{bmatrix} K_{(n-1)\times(n-1)}, K_{(n-1)\times 1} \\ K_{1\times(n-1)}, K_{1\times 1} \end{bmatrix}\right), K \equiv K_{RSGP}$$
2. Randomly sample $t^n_p$ from the inferred mean and variance
3. Update the reward-sensitive covariance, $K_{RS}$, and the full kernel, $K_{RSGP}$, based on $t^n_p$.
end

through searching the entire parameter space was inefficient and hindered convergence. Therefore, we implemented a two-step optimization. We first used the unrewarded trials to estimate $l_{SE}$ and $\sigma_{SE}$, and then used those fits to search for the best fit of the remaining hyperparameters ($l_{SE}$, $\sigma_{SE}$, $l_{RS}$, $\sigma_{RS}$, and $\sigma_0$) using all trials. The optimization of the multivariate likelihood function was achieved by line searching with quadratic and cubic polynomial approximations. The conjugate gradient was used to compute the search directions (*Rasmussen and Williams, 2006*). The landscape of likelihood indicated that the optimization was convex for a wide range of initial values (*Figure 5—figure supplement 1A*, *Table 2*).

The RSGP model fit to data provides a prediction of the distribution of $t_p$ on each trial based on previous trials ($t_p$ and reward history). We used this distribution to generate simulated values of $t_p$ for each session and repeated this process (n = 100) to estimate the distribution of $\mu(e^n)$ and $\sigma(e^n)$ in relation to $e^{n-1}$ using the same analysis we applied to behavioral data (*Figure 5C*). To derive an estimate of the slow part of error, $e_{\text{slow}}$, we first fitted the RSGP to the behavior, and then used the reduced RSGP that only included the slow kernel ($K_{SE}$) to predict the expected value of $e_{\text{slow}}$, that is the mean of a GP process governed by $K_{SE}$.

## Markov chain Monte Carlo (MCMC) model

The MCMC model was proposed by *Haith and Krakauer, 2014*. We adapted this algorithm to our task as shown in *Table 6*:

We have extensively explored the parameter space of this model. In simulations (*Figure 6C*, *Figure 6—figure supplement 1A*), we used 0.1\*$t_t$ and 0.05\*$t_t$ as the level of sampling noise **We** and execution noise **Wp** in step 2, respectively. The inverse temperature **β** was 100.

## Directed search (DS) model and characterizing the run-away behavior

The DS model is an RL-based learning scheme that sets the current gradient of the target interval based on the reward gradient in the two preceding trials. *Table 7* shows the underlying algorithm. We set the parameters of the model ($\alpha$, *We*, *Wp*) so as to minimize the MSE between monkeys' behavior and model prediction. With these parameters, the model often exhibited unstable run-away behavior. We defined run-away behavior as when the adaptive reward window reached a threshold ($t_t \pm 0.3\text{*}t_t$) and the number of consecutively unrewarded trials was larger than 20 trials (*Figure 6—figure supplement 1B*). With the same reward threshold, animals' behavior did not exhibit such run-away behavior.

## Relationship between neural activity and the slow component of behavior

We used linear regression to examine whether and to what extent the population neural activity could predict the slow component of error ($e_{\text{slow}}$) inferred from the RSGP model fits to behavior (as described in the previous section) using the following regression model:

$$e_{slow} = r\beta + \beta_0$$

where **r** represents a matrix ($nxN$) containing spike counts within a 250 ms of $N$ simultaneously recorded neurons across $n$ trials, $\beta_0$ is a constant, and **β** is an $N$-dimensional vector specifying the contribution of each neuron to $e_{\text{slow}}$. We used a random half of trials (training dataset) to find **β** and $\beta_0$ and the other half (validation dataset) to test the model, and quantify the success of the model

---

**Table 6.** Algorithm for generating time series based on MCMC model.

**for** each trial **do**
1. Keep an internal estimate of target $t_t\text{*}$ that is currently associated with the highest reward, $V(t_t\text{*})$.
2. On every trial, sample a new target, denoted $t_t^+$, from a Gaussian distribution with mean $t_t\text{*}$ and standard deviation **We**.
3. Generate $t_p$ by sampling from a Gaussian distribution with mean $t_t^+$ and standard deviation $t_t^+$. **Wp** (scalar noise). Assign the reward as the value of new sample $V(t_t^+)$.
4. Use a probabilistic Metropolis-Hastings rule to accept or reject $t_t^+$ as the new target depending on the relative values of $V(t_t^+)$, $V(t_t\text{*})$, and a free parameter **β** known in RL as the inverse temperature.

$$P(\text{Accept}) = \frac{e^{\beta \cdot V(t_t^+)}}{e^{\beta \cdot V(t_t^+)} + e^{\beta \cdot V(t_t\text{*})}}$$

**end**

**Table 7.** Algorithm for simulating the DS model.

**for** $n = 1...N$ **do**
1. Generate a new target estimation $t_t^n$ using gradients derived from the previous performance and reward
$\Delta t_t = \alpha \, (t_t^{n-1} - t_t^{n-2}) * (r^{n-1} - r^{n-2})$
$t_t^n = t_t^{n-1} + \Delta t_t + n_e$, in which the estimation noise $n_e \sim N(0, \mathbf{We})$
2. Generate $t_p^n$ with the added scalar production noise $\sim N(0, \mathbf{Wp})$
3. Compute the amplitude of reward $r^n$ based on $t_p^n$ and reward profile
**end**

by computing the Pearson correlation coefficient (**Figure 7B–D**) between the $e_{slow}$ inferred from RSGP model fits to behavior and $e_{slow}$ predicted from the neural data using the regression model.

We initially tested the regression model using spike counts immediately before Set and later extended the analysis to different time points throughout the trial. To do so, we aligned spike times to various events throughout the trial (Cue, Tar, Set, Go) and tested the regression model every 125 ms around each event (four time points after Cue, three time points before Tar, three time points after Tar, four time points before Set, four time points after Set and four time points before Go).

To ensure that the model was predictive and not simply overfitting noisy spike counts, we used a cross-validation procedure: for each session, we used a random half of the trials to estimate $\boldsymbol{\beta}_{Th}$, and the other half to quantify the extent to which $z_{Th}$ could predict $e_{slow}$. Note that some correlations are negative because of cross-validation (we used a random half of data to estimate the drift direction and the other half for estimation correlations). Otherwise, all correlations should have been non-negative. Note that the number of sessions was combined across all four trial types as shown in **Figure 7B–D** bottom.

## Statistical analysis

Mean ± standard deviation (s.d.), Mean ± standard error of the mean (SEM) or median ± median absolute deviation (MAD) were used to report statistics. We detailed all the statistics in the Results and figure captions. All hypotheses were tested at a significance level of 0.01 and p-values were reported. We used $t$-tests to perform statistical tests on the following variables: (1) weber fraction (ratio of standard deviation to mean of $t_p$), (2) cross correlation between pairs of trials of different lag and trial type, (3) modulation of the variability by discrete or graded reward (4) variance terms in the RSGP model ($\sigma^2_{SE}$, $\sigma^2_{RS}$, and $\sigma^2_0$) which were assumed to be normally distributed. We used one-tailed paired, two-tailed paired or two-sample $t$-tests depending on the nature of data and question. The length scale parameters of the RSGP model ($l_{SE}$ and $l_{RS}$) were not normally distributed. Therefore, we used a one-way ANOVA to test whether the two were significantly different. We used a two-sample $F$-test to compare the variability of production intervals for different pair-trial conditions (H0: equal variance).

## Mathematical notation

| Symbol | Description |
|---|---|
| $t_p^n$ | Production time in the n-th trial |
| $e^n$ | $(t_p^n - t_t)/t_t$, relative error of production time in the n-th trial |
| $\mu(e)$ | Mean of relative error |
| $\sigma(e)$ | Standard deviation of relative error |
| $K(i,j)$ | Covariance between trial $i$ and trial $j$ |
| $\sigma^2$ | Signal variance or noise variance associated with a Gaussian process |
| $l$ | The length scale of the squared-exponential covariance function |
| $r^n(t)$ | Population spiking at time t in the n-th trial |
| $\beta$ | Regression coefficient |
| $z$ | Projection of population spiking activity onto a low dimensional representation |

## Acknowledgements

MJ is supported by NIH (NINDS-NS078127), the Sloan Foundation, the Klingenstein Foundation, the Simons Foundation, the McKnight Foundation, and the McGovern Institute. NM is supported by the Center for Sensorimotor Neural Engineering.

## Additional information

### Funding

| Funder | Grant reference number | Author |
|---|---|---|
| National Institute of Neurological Disorders and Stroke | NINDS-NS078127 | Mehrdad Jazayeri |
| Simons Foundation | 325542 | Mehrdad Jazayeri |
| Simons Foundation | 542993SPI | Mehrdad Jazayeri |
| McKnight Endowment Fund for Neuroscience | | Mehrdad Jazayeri |
| Esther A. and Joseph Klingenstein Fund | | Mehrdad Jazayeri |

The funders had no role in study design, data collection and interpretation, or the decision to submit the work for publication.

### Author contributions

Jing Wang, Conceptualization, Data curation, Software, Formal analysis, Validation, Visualization, Writing - review and editing; Eghbal Hosseini, Conceptualization, Data curation, Formal analysis, Validation, Investigation, Visualization, Writing - review and editing; Nicolas Meirhaeghe, Data curation, Formal analysis, Writing - review and editing; Adam Akkad, Data curation, Writing - review and editing; Mehrdad Jazayeri, Conceptualization, Supervision, Funding acquisition, Investigation, Writing - original draft, Project administration, Writing - review and editing

### Author ORCIDs

Jing Wang https://orcid.org/0000-0002-4761-6748
Eghbal Hosseini http://orcid.org/0000-0002-0088-9765
Nicolas Meirhaeghe http://orcid.org/0000-0002-3785-0696
Mehrdad Jazayeri https://orcid.org/0000-0002-9764-6961

### Ethics

Human subjects: The Committee on the Use of Humans as Experimental Subjects at the Massachusetts Institute of Technology approved the human experiments (Protocol Number: 1304005676). The Committee on the Use of Humans as Experimental Subjects at Massachusetts Institute of Technology approved all human experiments. As per our approved protocol, all human participants provided consent for the use and publication of data prior to data collection.

Animal experimentation: All procedures conformed to the guidelines of the National Institutes of Health. All animals were handled according to our protocol (Protocol Number: 0119-002-22) that was approved by the Committee of Animal Care at the Massachusetts Institute of Technology.

### Decision letter and Author response

Decision letter https://doi.org/10.7554/eLife.55872.sa1
Author response https://doi.org/10.7554/eLife.55872.sa2

## Additional files

### Supplementary files

- Transparent reporting form

## Data availability

Matlab codes for RSGP simulation and model fitting are available at https://github.com/wangjing0/RSGP. Data are avaible at https://jazlab.org/resources/.

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
