## [Decision Letter]

**Acceptance summary:**

How variability in motor actions changes with learning is not well understood, and the area is waiting for advances in both computational theorization and related neural underpinnings. The present study contributes by investigating a motor timing task in which reward-dependent learning and timing variability interact. Importantly, the observed behavioral signatures enable new modeling of motor reinforcement learning and characterizing the underlying neural substrate in the cortico-basal ganglia circuit.

**Decision letter after peer review:**

Thank you for submitting your article "Reinforcement regulates timing variability in thalamus" for consideration by *eLife*. Your article has been reviewed by three peer reviewers, including Kunlin Wei as the Reviewing Editor and Reviewer #1, and the evaluation has been overseen by Michael Frank as the Senior Editor. The following individual involved in review of your submission has agreed to reveal their identity: Bruno B Averbeck (Reviewer #2).

The reviewers have discussed the reviews with one another, and the Reviewing Editor has drafted this decision to help you prepare a revised submission.

As the editors have judged that your manuscript is of interest, but as described below that additional experiments/analyses are required before it is published, we would like to draw your attention to changes in our revision policy that we have made in response to COVID-19 (https://elifesciences.org/articles/57162). First, because many researchers have temporarily lost access to the labs, we will give authors as much time as they need to submit revised manuscripts. We are also offering, if you choose, to post the manuscript to bioRxiv (if it is not already there) along with this decision letter and a formal designation that the manuscript is "in revision at *eLife*". Please let us know if you would like to pursue this option. (If your work is more suitable for medRxiv, you will need to post the preprint yourself, as the mechanisms for us to do so are still in development.)

Summary:

How variability in motor actions changes with learning is not well understood. The present study starts off with a motor timing task where monkeys and human participants were required to produce timing intervals on the scale of hundreds of milliseconds. Two behavioral features emerge: one is strong correlations of timing produced intervals within effectors and intervals, the other is increased timing variability when the produced intervals were away from the target or mean interval. The first feature is interpreted as a slow drift of timing memory, and the second feature is interpreted as from strategic use of exploratory variance to guard against undesirable variability, i.e., the memory drift. The study provided a computational model to incorporate continuous rewards in the framework of reinforcement learning. Furthermore, the study also correlated the direct recording from the thalamus, DMFC, and caudate to the two behavioral findings, and found that the thalamus showed reward-dependent neural activities with clear effector-specificity. The authors conclude that the nervous system strategically regulates the variability based on reinforcement to reduce the detrimental effect of undesirable variability in the system in the exploration-exploitation framework.

There was agreement among the reviewers that these results will be of interest to the audience of *eLife*. However, there are critical issues that need to be addressed before the paper being considered for acceptance. The first major concern is essential, given that it is about whether the current data can be interpreted as the way it is.

Essential revisions:

1) The main message of the paper is to explain the non-stationarity in the timing variance in the exploration-exploitation framework, but this is questionable. As the reward follows a specific function of timing error, the first derivative of the reward feedback could effectively guide the trial-by-trial modulation of timing behaviors. For example, a decrease in reward would signal a departure from the target interval, which can be used to guide an appropriate response in the next trial. In this case, there is no need to crank up noise with a decrease in reward, as the exploration-exploitation framework would predict. Thus, the observed changes in variance can be explained by trial-by-trial learning based on the explicit reinforcement feedback signal, without invoking the idea of random exploration as in the exploration-exploitation framework. Recent theoretical approaches to model exploration-exploitation behaviors have emphasized both random vs. directed exploration (Wilson et al., 2014), but the current study appears to assume that all exploration should be random. Considering that the task has a 1-D continuous reward function, directed exploration is well possible. This is a critical question given that the main implication of the study is about "…the nervous system makes strategic use of exploratory variance…". In fact, the whole paper is framed as probing reinforcement-guided exploration as opposed to trial-by-trial supervised learning.

In a similar vein, the non-stationarity in the variance is caused by the reward magnitude (the specific reward function used here), not necessarily a refutation of stationarity of interval time. It has been acknowledged by the author that this is not a rejection of the interval timing model, but the paper continues to imply it in the Abstract and in the Results.

Did the subjects not use information from the first derivative of the reward to update their produced intervals? It is not even clear that how many produced intervals fell within the rewarded range, and how many were simply unrewarded. The details of the reward magnitude and the monkey's behavioural adaptations to these changes in reward need to be clarified. To make the original claims of the paper hold, the authors need to clarify whether the results can be explained by simple trial-by-trial adjustments based on the first derivative of the reward function.

2) The human task with a probabilistic reward has not been directly compared with the monkey experiment, though both are displayed in Figure 5. Related to question 1), can the findings with probabilistic rewards suffice to rule out the possibility that the first derivative of reward feedback is the driving force for the observed variance changes?

3) The asymmetry of interval variance was evident for both monkeys when the target interval of 1500ms was produced (Figure 1D), but it was left unexplained. This asymmetry shows a much higher variance for the shorter intervals, i.e., skewed to the 800ms target interval. Was this caused by a trivial fact that the monkeys were producing the wrong interval? The data from Monkey D (Figure 1B) appear to suggest this is possible (a few intermediate intervals with the 1500ms target interval). This can be further verified by raw data from Monkey A, which is currently absent in Figure 1. Furthermore, if you use decoding on the neural activity, can you predict whether the monkeys are indeed trying to produce long intervals, in the trials in which they produce long intervals that are too short?

4) Figure 8—figure supplement 6 shows that speed variability differed between rewarded and unrewarded trials for DMPC and caudate, but not for the thalamus. Does this contradict the implied role of the thalamus in reinforcement learning?

5) The study highlights the role of the thalamus in reward-based learning; recent studies have hypothesized that fronto-thalamic circuits are necessary for quick processing of complex task-relevant information. Given the task investigated here is also complex (associating reward size to task performance across trials), it is well possible that several areas of frontal cortex are also involved. How to link the current findings to the hypotheses of fronto-thalamic circuits?

---

## [Author Response]

Essential revisions:1) The main message of the paper is to explain the non-stationarity in the timing variance in the exploration-exploitation framework, but this is questionable. As the reward follows a specific function of timing error, the first derivative of the reward feedback could effectively guide the trial-by-trial modulation of timing behaviors. For example, a decrease in reward would signal a departure from the target interval, which can be used to guide an appropriate response in the next trial. In this case, there is no need to crank up noise with a decrease in reward, as the exploration-exploitation framework would predict. Thus, the observed changes in variance can be explained by trial-by-trial learning based on the explicit reinforcement feedback signal, without invoking the idea of random exploration as in the exploration-exploitation framework. Recent theoretical approaches to model exploration-exploitation behaviors have emphasized both random vs. directed exploration (Wilson et al., 2014), but the current study appears to assume that all exploration should be random. Considering that the task has a 1-D continuous reward function, directed exploration is well possible. This is a critical question given that the main implication of the study is about "…the nervous system makes strategic use of exploratory variance…". In fact, the whole paper is framed as probing reinforcement-guided exploration as opposed to trial-by-trial supervised learning.

The reviewers are concerned that the observed behavioral effects may be due to directed (as opposed to random) exploration. We are grateful for this comment as it reveals that the emphasis we put on variability caused a serious misunderstanding of the mechanism we proposed. Our findings do not support the notion that the brain simply makes behavior noisy. Indeed, the strategy implemented by our proposed Reward Sensitive Gaussian Process (RSGP) is a combination of directed and random explorations. However, since our original manuscript failed to communicate this point, we have made several revisions throughout the revised manuscript to address this problem.

The first major revision we made was to add an entire new section and a new figure (Figure 6) where we compare our proposed RSGP model to alternative RL models. In that section, we considered two alternatives, one model that implements an RL sampling algorithm (MCMC, which is more similar to random exploration), and the other, which we refer to as the directed search (DS) model, implements the algorithm suggested by the reviewers: it compares reward in the last two trials; when reward increases, the model moves the estimate in the same direction, and when reward decreases, the model reverses direction. We have provided a detailed explanation of this model in the paper. Both models failed to capture the full gamut of behavioral observations in humans and monkeys. Here, we summarize the failure points of the DS model that was proposed by the reviewers. First, for the same level of behavioral variability and the same reward rate, the DS exhibited runaway behavior (i.e., the model’s output could move arbitrarily far from the target interval). Second, for the subset of simulated sessions that the behavior remained stable, DS was unable to capture the long-term correlations in behavioral responses present in the data. Third, and perhaps most importantly, DS was unable to exhibit the characteristic U-shaped relationship between variability and preceding error. Therefore, the behavior cannot be explained by the DS model.

The second major revision we made was to include a new section titled “Directed versus random exploration.” In this new section, we highlighted the importance of both directed and random explorations in our own data. The point that we failed to communicate in the original manuscript was that RSGP is *not* simply adjusting the variance; RSGP implements Bayesian inference over the desired interval and thus dynamically adjusts both the bias and variance. We noted this point in the original manuscript (“GPs offer a nonparametric Bayesian fit to long-term serial correlations”) but should have gone into further detail, which we have done in this new section. The key section added to the revised manuscript is included below for convenience:

“To highlight the behavioral characteristics of the RSGP model, it is useful to compare its behavior to that of the MCMC and DS models. MCMC and DS can be viewed as autoregressive models whose coefficients depend on past rewards. The MCMC sets all coefficients to zero except a single trial in the past returned by the Metropolis-Hastings sampling algorithm. The DS sets all coefficients to zero except the last two that are determined based on the corresponding reward gradient and a fixed learning rate. The RSGP can also be written in terms of an autoregressive model with nonstationary reward-dependent coefficients. However, the key feature that distinguishes the RSGP is that it performs Bayesian inference over *t*__. The covariance function (or kernel) of the Gaussian process defines the coefficients and acts as a prior for future samples. In the RSGP, the addition of a reward-sensitive kernel allows the coefficients to be updated continuously based on the presence or absence of reward. When the reward rate is high, RSGP implements a strategy that is akin to directed exploration: it increases its reliance on the prior and drives responses toward previously rewarded trials. In contrast, when the reward rate is low, the RSGP relies more on random explorations: it generates samples from a wider distribution. Therefore, RSGP strikes a balance (in the Bayesian sense) between bias (directed exploration) and variance (random exploration) as needed by the history of outcomes.”

Finally, we revised the Discussion to make sure that the importance of both directed and random explorations, as prescribed by the RSGP model, are highlighted. The revised text in the Discussion is copied here for convenience: “A key challenge in the RL literature is to distinguish between conditions in which reinforcement motivates directed versus random explorations. […] Indeed, the RSGP suggests that the distinction between directed and random explorations is somewhat arbitrary, and advocates an alternative viewpoint in which the effect of reinforcement is quantified in terms of how it alters the full distribution of future responses (e.g., both the mean and variance).”

In a similar vein, the non-stationarity in the variance is caused by the reward magnitude (the specific reward function used here), not necessarily a refutation of stationarity of interval time. It has been acknowledged by the author that this is not a rejection of the interval timing model, but the paper continues to imply it in the Abstract and in the Results.

We respectfully disagree. Despite the assumption of stationarity in dominant models of scalar timing, motor timing is inherently non-stationary. There is a rich and extensive literature showing this nonstationarity. These fluctuations have been reported in many tasks (Weiss, Coleman and Green, 1955; Merrill and Bennett, 1956; Gilden, Thornton and Mallon, 1995) and can be relatively strong in movements (Chaisanguanthum, Shen and Sabes, 2014), reaction times (Laming, 1979), and interval timing (Chen, Ding and Kelso, 1997; Murakami et al., 2017). In fact, we do not know of any paper that has shown stationary behavior associated with time interval production from memory. Therefore, the presence of nonstationary motor timing is not controversial (even though dominant models of interval timing do not account for it).

We are not the first to report these nonstationarities, but, to our knowledge, our work is the first attempt to dissect the factors that contribute to these nonstationarities. Our results identify two contributing factors: memory drift and trial outcome. Our results do not reject scalar variability but it seeks to explain its underlying biological factors. We have revised the text in the relevant section of the Results further stressing the ubiquity of non-stationarity in memory-based motor timing tasks.

Did the subjects not use information from the first derivative of the reward to update their produced intervals? It is not even clear that how many produced intervals fell within the rewarded range, and how many were simply unrewarded. The details of the reward magnitude and the monkey's behavioural adaptations to these changes in reward need to be clarified.

The reviewers note that “It is not even clear how many produced intervals fell within the rewarded range, and how many were simply unrewarded.” We apologize for not providing this information more coherently. To address this shortcoming, we have added two new sections in the Materials and methods titled “Reward function for monkeys in the main task” and “Reward function for humans in the main task” where we provide a detailed description of reward contingencies. For example, the section related to animals is as follows:

“The reward was determined by a truncated triangular function of error (*t**p*__*-t**t*__). […] The auditory feedback was one of the computer’s default tones.”

Additionally, we highlighted some of the key points that were cause for confusion in the Results section (e.g., the use of one-up-on-down staircase procedure), as follows:

“Reward was provided when the relative error, defined as *e* = (*t**p*-*t**t*)/*t**t* was within an experimentally-controlled acceptance window. On rewarded trials, the magnitude of reward decreased linearly with the size of the error. The width of the acceptance window was controlled independently for each trial type using a one-up-one-down staircase procedure (see Materials and methods) so that animals received reward on nearly half of the trials (Figure 1C, inset).”

To make the original claims of the paper hold, the authors need to clarify whether the results can be explained by simple trial-by-trial adjustments based on the first derivative of the reward function.

The reviewers ask that we “clarify whether the results can be explained by simple trial-by-trial adjustments based on the first derivative of the reward function.” We can think of two potential interpretations for “the first derivative of the reward function”. One possible interpretation is the change of reward across preceding trials (e.g., the difference of reward in the last two trials). This is exactly the idea behind the alternative DS model that we have tested in the revised manuscript, and discussed in our response to the first major comment (above).

An alternative interpretation of “the first derivative” is the derivative of the triangular function we used to determine reward magnitude. However, we doubt this is what the reviewers have in mind as this does not provide a coherent explanation of the results. First, the human experiments did not involve a triangular reward function, and yet, behavioral characteristics were similar. Second, it is impossible to infer the derivative of a function from a single sample (e.g., single trial). For example, if on a given trial, the animal receives half the maximum reward, the produced interval could be either slightly longer or slightly shorter than the target interval (due to the symmetry of the reward function), and the animal cannot know which is the case unless it relies on multiple trials (which brings us back to the first interpretation). Third, we found a systematic increase in variability even for the consecutive unrewarded trials (i.e., responses that fall outside the reward triangle) for which the derivative is zero. In sum, we think it is not possible to infer the derivative of a symmetric reward function solely based on the magnitude of the reward in a single trial, and thus the only way to rely on the reward gradient is to rely on more than one preceding trial. In the revised manuscript, we have built such a model (the DS model) and shown that it fails to capture several key features of behavior including the U-shaped relationship between variability and the preceding error. We have also explained that our proposed RSGP model implements a similar strategy but does not suffer from the shortcomings of the DS model because it integrates information on a longer timescale (more than the last two trials).

2) The human task with a probabilistic reward has not been directly compared with the monkey experiment, though both are displayed in Figure 5. Related to question 1), can the findings with probabilistic rewards suffice to rule out the possibility that the first derivative of reward feedback is the driving force for the observed variance changes?

In the original submission, Figure 5 showed the RSGP model fits to behavior in the main experiment – not the probabilistic reward experiment. The results for the probabilistic reward experiment was shown in Figure 4, which did not include any animal experiments. Therefore, we are unsure about which figure and what aspect of the results is being referenced.

However, we understand the more general question about whether the probabilistic reward experiment can be used to test the DS model. In our response to major comment 1, we listed the multiple reasons why the DS model cannot explain the behavioral results (e.g., does not lead to the U-shaped relationship of variability as a function of the preceding error). In addition, the directed search strategy can move in the wrong direction when the rewards are probabilistic (e.g., when a large error is rewarded by chance). As the reviewers have surmised, the probabilistic reward experiment strengthens the conclusion because the directed search, unlike RSGP, only acts on the mean and not the variance, and thus cannot fully capture the behavioral observation (Figure 5—figure supplement 2).

3) The asymmetry of interval variance was evident for both monkeys when the target interval of 1500ms was produced (Figure 1D), but it was left unexplained. This asymmetry shows a much higher variance for the shorter intervals, i.e., skewed to the 800ms target interval. Was this caused by a trivial fact that the monkeys were producing the wrong interval? The data from Monkey D (Figure 1B) appear to suggest this is possible (a few intermediate intervals with the 1500ms target interval). This can be further verified by raw data from Monkey A, which is currently absent in Figure 1.

The reviewers point to an observation in Figure 1D that the interval variance was asymmetric for the 1500 ms target interval. This asymmetry is due to a number of inconsequential aspects of the data. First, the distribution of produced intervals with respect to the target 1500 ms interval was shifted to the left of the target 1500 ms (Monkey A: -11.0 ms for Eye-Long and -4.3 ms for Hand-Long; Monkey D: -45.3 ms for Eye-Long, and -11.9 ms for Hand-Long). This caused a sampling bias in the density and magnitude of errors on the two sides of the curve. Second, because of this bias, we were able to estimate the curve further out to the left of the 1500 ms (more bins on the short side), which made the curve appear more asymmetric. Third, the presence of scalar variability causes a slight skew in the distribution of produced intervals that further skews the produced interval distribution toward shorter intervals and intensifies the asymmetry. Since these details are not relevant to our main hypothesis, we removed this panel from Figure 1 and replaced it with an example behavioral session from Monkey A as requested by the reviewers.

However, as part of this comment, the reviewers express concern that perhaps monkeys made categorical mistakes between the Short and Long trial types. To address this concern, we performed a detailed analysis of the produced intervals asking whether animals mixed up between the two target intervals (“monkeys were producing the wrong interval”). To do so, we applied a Gaussian mixture model (GMM) to the full distribution of produced intervals (for both 800 and 1500 ms) and asked whether any of the produced intervals were in the wrong mode with respect to the desired interval. Results indicated that the animals’ behavior was highly stable across trials and sessions and made nearly no categorical mistake between trial conditions. We have clarified this point in the relevant section of the Results: “There were no errors associated with using the wrong effector, and the number of errors with respect to the target interval were extremely small (~0.79% misclassified trials based on fits to a Gaussian mixture model).”

Furthermore, if you use decoding on the neural activity, can you predict whether the monkeys are indeed trying to produce long intervals, in the trials in which they produce long intervals that are too short?

The reviewers also asked that we use a neural classifier to determine which of the two intervals the animals’ were aiming to produce. We generally are not in favor of this approach, which assumes that a neural classifier of behavior is a more reliable readout of behavior than the behavior itself. However, for completeness, we performed the suggested analysis. We built a classifier of interval type (Short or Long) based on population spiking data. We used the trials that were correctly classified by the GMM for training the classifier, and asked whether the trials that were misclassified by the GMM were also misclassified by the neural classifier. The performance of the classifier was at chance level (55+/-27% in the thalamus, 45+/-31% in DMFC and 60+/-23% in caudate).

4) Figure 8—figure supplement 6 shows that speed variability differed between rewarded and unrewarded trials for DMPC and caudate, but not for the thalamus. Does this contradict the implied role of the thalamus in reinforcement learning?

We apologize for the lack of clarity. On the contrary, this is a direct prediction of our prior work (Wang et al., 2018). Previously, we found the neurons in the thalamus encode the desired interval by adjusting their tonic firing rate, and not by changing the speed at which firing rates changed over time. In contrast, we found that downstream neurons in DMFC and caudate were largely explained in terms of changing speed. The current results are consistent with and validate those findings.

In the original manuscript, we had provided an explanation of this point in the Discussion as follows: “Moreover, it has been shown that the nature of signals in DMFC-projecting thalamus and DMFC during motor timing are indeed different: DMFC neurons have highly heterogeneous response profiles that evolved at different speeds depending on the interval, whereas thalamic neurons carried signals whose strength (i.e., average firing rate) encoded the underlying speed.”

We then went on to explain how this previous finding explains the presence of speed effects in DMFC and not thalamus. We have clarified this in the text as follows: “As predicted by our hypothesis, the effect of reward on neural activity in the thalamus was different from that in the DMFC and caudate. […] These results further substantiate our hypothesis that reward regulates variability by adjusting the average firing rates in thalamus, and that this effect leads to the control of the variance of the speed at which neural trajectories evolve in the DMFC and caudate.”

5) The study highlights the role of the thalamus in reward-based learning; recent studies have hypothesized that fronto-thalamic circuits are necessary for quick processing of complex task-relevant information. Given the task investigated here is also complex (associating reward size to task performance across trials), it is well possible that several areas of frontal cortex are also involved. How to link the current findings to the hypotheses of fronto-thalamic circuits?

We may have misunderstood this comment but it appears that this point was already addressed extensively in two parts of the Discussion in the original manuscript. We have made some minor revisions to those sections with the hope that the point is now more clearly communicated. We have included these sections here for the reviewers’ convenience.

First section: “we cannot rule out the possibility that behavioral control is mediated by other patterns of activity in the thalamus as well as DMFC. Indeed, with a complementary analysis using an unconstrained decoding approach, we were able to find patterns of activity in all three brain areas that simultaneously reflected the effects of memory drift (Figure 7) and reinforcement (Figure 8—figure supplement 3).”

Second section: “While we do not know which areas might directly control motor timing variability, we note that the area of thalamus we have recorded from receives information from three major sources, the frontal cortex, the output nuclei of the basal ganglia, and the deep nuclei of the cerebellum (Middleton and Strick, 2000; Kunimatsu et al., 2018). […] To act as an effective learning mechanism, such correlated variability must be additionally sensitive to reward-dependent neuromodulatory signals such as dopamine (Frank et al., 2009) possibly by acting on local inhibitory neurons (Huang et al., 2019).”

We hope that these Discussion points (which are now further revised to clarify the key points) address the reviewers’ comment.